# Decoding Methods in Neural Language Generation: A Survey

**Sina Zarrieß [1],\* , Henrik Voigt [2] and Simeon Schüz [1]**

[1] Faculty for Linguistics and Literature Studies, Bielefeld University, 33615 Bielefeld, Germany; simeon.schuez@uni-bielefeld.de

[2] Faculty of Mathematics and Computer Science, Friedrich Schiller University Jena, 07743 Jena, Germany; henrik.voigt@uni-jena.de

\* Correspondence: sina.zarriess@uni-bielefeld.de

**Abstract:** Neural encoder-decoder models for language generation can be trained to predict words directly from linguistic or non-linguistic inputs. When generating with these so-called end-to-end models, however, the NLG system needs an additional decoding procedure that determines the output sequence, given the infinite search space over potential sequences that could be generated with the given vocabulary. This survey paper provides an overview of the different ways of implementing decoding on top of neural network-based generation models. Research into decoding has become a real trend in the area of neural language generation, and numerous recent papers have shown that the choice of decoding method has a considerable impact on the quality and various linguistic properties of the generation output of a neural NLG system. This survey aims to contribute to a more systematic understanding of decoding methods across different areas of neural NLG. We group the reviewed methods with respect to the broad type of objective that they optimize in the generation of the sequence—likelihood, diversity, and task-specific linguistic constraints or goals—and discuss their respective strengths and weaknesses.

**Keywords:** neural language generation; decoding; beam search; sampling; diversity

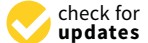



## 1. Introduction

The rise of deep learning techniques in NLP has significantly changed the way natural language generation (NLG) systems are designed, developed, and trained with data. Traditional, rule- or corpus-based NLG systems typically modeled decisions at different levels of linguistics processing in an explicit and symbolic fashion [1,2].

In contrast, recent neural network architectures for generation can be trained to predict words directly from linguistic inputs or non-linguistic data, such as database records or images. For this reason, neural generators are commonly referred to as "end-to-end systems" [3–7].

It is less commonly noticed, however, that the neural end-to-end approach to generation is restricted to modeling *word* probability distributions, whereas the step of determining the output *sequence* is not handled in the model itself. Thus, neural NLG system generally need an additional *decoding* procedure that operates in symbolic space and defines how words are strung together to form sentences and texts. This survey focuses on the decoding stage in the neural language generation process. It provides an overview of the vast range of decoding methods that have been developed and used in recent research on neural language generation and discusses their respective strengths and weaknesses.

The development of a neural architecture for generation involves many steps and aspects, starting from the definition of the task, the collection and preparation of data, the design of the model and its training, and the evaluation. Recent surveys in NLG cover these aspects very well but do not address the topic of decoding in particular, e.g., Gatt and Krahmer [2]'s very extensive survey on different NLG tasks, architectures, and evaluation methods. Similarly, in many recent papers on NLG systems or tasks, the decoding method

does not play a central role. Often, it is reported as a technical detail of the experimental set-up. At the same time, research into decoding has become a real trend in the area of neural language generation. Numerous papers have been published in recent years showing that the choice of decoding method has a considerable impact on the quality and various linguistic properties of the generation output. This survey aims to make this trend more visible and to contribute to a more systematic understanding of decoding and its importance for neural NLG.

### 1.1. Motivation and Overview

In a neural NLG system, the decoding method defines the way the system handles its search space over potential output utterances when generating a sequence. Generally, in neural language generation, this search space is infinite, i.e., it grows exponentially with the length of the output sequence. Therefore, the decoding procedure is an important part of the neural NLG pipeline where non-trivial design decisions are taken by the developer of the NLG system. The first goal of this survey is to introduce the notion of decoding and show its importance for different neural (and non-neural) NLG frameworks (Section 2).

The most well-known and de-facto standard decoding procedure in NLG is beam search, a general and traditional search algorithm which dates back to Lowerre [8]'s work on speech recognition. Since the advent of neural NLG, however, researchers have noticed shortcomings of beam search and its many variants that are used more or less systematically in practice. The second goal of this survey is to provide an in-depth overview of definitions and analyses of beam search in neural NLG (Section 3).

While beam search is designed to maximize the likelihood of the generated sequence, many recently developed decoding methods prioritize other generation objectives. Most notably, a considerable body of work has investigated decoding methods that increase the so-called "diversity" of generation output. Section 4 introduces different notions of diversity used in the decoding literature and reviews the corresponding methods.

While likelihood-oriented and diversity-oriented decoding is rather task-independent, other lines of work have investigated decoding methods that explicitly introduce task-specific objectives and linguistic constraints into the generation process. Modeling constraints that control the behavior of an NLG system for particular tasks or situations is a notorious problem in neural NLG, given the complex black-box design of neural network architectures. Decoding seems to offer an attractive solution (or work-around) to this problem as it operates on the symbolic search space representing generation candidates. Section 5 will summarize works that view decoding as a means of controlling and constraining the linguistic properties of neural NLG output.

Finally, the decoding methods reviewed in this survey do not only constitute interesting algorithms on their own, since they are closely connected to general themes and questions that revolve around neural NLG. As the above overview has already shown, decoding methods show important differences with respect to their objectives and underlying assumptions of the generation process. Section 6 provides some discussion of the challenges and open questions that are brought up by decoding, but concern neural NLG in general.

In short, the goals of this survey can be summarized as follows:

- overview decoding across different neural NLG frameworks (Section 2),
- review of different variants of beam search-based decoding and summarize the debate about strengths and weaknesses of beam search (Section 3),
- discuss different notions of diversity in the decoding literature and summarize work on diversity-oriented decoding methods (Section 4),
- summarize work on task-specific decoding (Section 5), and
- discuss challenges in neural NLG brought up by work on decoding (Section 6).

*1.2. Scope and Methodology*

This survey focuses on decoding methods for neural NLG-but how do we define NLG in the first place? A very popular definition of NLG is the one by Reiter and Dale [9], which states that NLG is "is concerned with the construction of computer systems than can produce understandable texts in English or other human languages from some underlying non-linguistic representation of information". In recent years, however, the research questions and modeling approaches in NLG overlap more and more with questions addressed in areas, such as text-to-text generation (e.g., summarization) [10,11], machine translation [10], dialog modeling [12], or, notably, language modeling [11]. Here, the input is not necessarily language-external data but linguistic input, e.g., text. Gatt and Krahmer [2]'s survey focuses mostly on "core" NLG where the input to the system is non-linguistic. In our discussion of decoding, we will see that this distinction is very difficult to maintain as methods for neural data-to-text generation are often directly inspired by and compared to methods from other areas subsumed under or related to NLG, particularly in machine translation and language modeling. Hence, in most of this article, we will adopt a rather loose definition of NLG and report on decoding methods used to generate text in neural encoder-decoder frameworks.

This survey aims at a comprehensive overview of different approaches to decoding and their analysis in the recent literature. Therefore, it includes a diverse set of papers published at major international NLP, ML, and AI venues since the development neural NLG in 2015, i.e., papers that introduce particular decoding methods, that present analyses of decoding, or that report relevant experiments on decoding as part of a particular NLG system. This survey also includes papers published before the advent of neural NLG, introducing foundational work on decoding methods that are still widely used in neural NLG. Furthermore, this survey provides a perspective on decoding methods from a practical perspective. We have compiled a list of papers on well-known NLG systems spanning the different NLG tasks just discussed and report their decoding method, even if it is not central in that paper. Tables 2 and 3 show this list, which contains systems that either implement a decoding method relevant for this survey or constitute a popular approach in their sub-area according to their number of citations and publication venue. Table 2 summarizes the text-to-text generation systems which process linguistic inputs, whereas Table 3 lists data-to-text systems that take non-linguistic data as input. This takes up the distinction between different types of NLG tasks discussed above and allows for a comparison between these overlapping areas.

## 2. Decoding Across NLG Frameworks and Tasks

This survey is devoted to decoding methods that are defined as inference procedures external to the neural NLG model and that can be used broadly and independently across different NLG tasks and architectures. Hence, most of this survey will abstract away from the inner workings of neural NLG architectures and models. At the same time, we will also see that many decoding methods are designed to address particular shortcomings of neural generation systems and challenges that arise in the neural encoder-decoder generation framework. Therefore, before going into the details of decoding methods in the remainder of this article, this section will briefly introduce some basic NLG frameworks and discuss why and where they require a decoding procedure. We will start with pre-neural statistical NLG systems in Section 2.1, move on to autoregressive neural generation in Section 2.2 and non-autoregressive models in Section 2.3. Section 1.2 gives an overview of different NLG tasks considered in this survey.

*2.1. Pre-Neural NLG*

First of all, template- or rule-based approaches constitute an important type of NLG system that is often relevant in practical applications. These systems offer "hand-built" solutions for specific generation domains or specific parts of a generation pipeline [13–16] and can be designed at varying levels of linguistic complexity [17]. Generally, they explicitly

restrict the system's search space to a finite set of utterances, use rules to fill in pre-specified utterance templates, and do not require a decoding method.

Other NLG frameworks have integrated grammar-based components into hybrid architectures that leveraged a statistical component to rank or score decisions specified by the grammar. Early approaches in corpus-based NLG followed a generate-and-rank approach where a (more or less sophisticated) grammar was used to produce an exhaustive set of generation candidates which was subsequently ranked globally by a language model or some other type of scoring or reranking model [18–21]. These systems deal with a larger search space than fully template- or rule-based systems, but still search the entire, finite hypothesis space for the globally optimal output candidate.

Subsequent work on stochastic language generation aimed at methods which avoid an exhaustive ranking or traversal of the candidate space. Another way to integrate grammar-based generation with statistical decision making was introduced in the probabilistic CFG-based generator by Belz [22]. Their system was built for the task of weather forecast generation and features expansion rules with weights or probabilities learned from a corpus. Belz [22] experiment with three decoding strategies for searching the space of possible expansions: greedy search, viterbi search and greedy roulette-wheel generation. The latter two correspond to two main types of decoding methods discussed in this survey, i.e., viterbi search as a search-based decoding method and roulette-wheel generation as a sampling-based method that favors diversity. Belz [22]'s experiments showed that greedy search outperformed the other decoding methods.

Subsequent and concurrent work on statistical NLG has aimed at implementing generation models that do not require a grammar as a backbone and can be learned in an end-to-end fashion and trained directly on input-output pairs. Angeli et al. [23] present a simple, domain-independent method for training a generator on different data-to-text generation corpora that align sentences to database records. Their system decomposes the generation process into a sequence of local content selection and realization decisions, which are handled by discriminative classifiers. Thus, in contrast to recent, neural end-to-end systems (see Section 2.2), their model does not directly predict words from a given input, but implements an intermediate level of processing that models the structure of the output sequence. Angeli et al. [23] discuss the possibility to use different decoding methods, i.e., greedy search, sampling and beam search, but state that greedy search outperformed beam search in their setting.

In a similar vein, Konstas and Lapata [24] present an approach to concept-to-text generation which they call unsupervised as it does not assume explicit alignments between input representations (database records) and output text. Their framework is based on a basic probabilistic CFG that captures the syntactic relations between database records, fields, and words. Importantly, their system represents the search space as a set of trees encoded in a hyper-graph. A core component of their system is a relatively advanced decoding method as a naive traversal of the hyper-graph would be infeasible. For decoding, they adopt cube-pruning [25], a variant of beam-search for syntax-based machine translation which allows them to interleave search with language model scoring. Mairesse and Young [26] developed the BAGEL system as a fully stochastic approach for generation in a dialog system setting. Their approach does not rely on a hand-coded grammar, but frames the generation task as a search over Factored Language Models. These can be thought of as dynamic Bayesian networks and constitute, according to Reference [27], a principled way of modeling word prediction in a large search space. Thus, in BAGEL, the language generator's task is to predict, order and realize a sequence of so-called semantic stacks (similar to slots). A core component of BAGEL is a decoding procedure that divides the search problem into three sequential sub-tasks, i.e., the ordering of mandatory stacks, the prediction of the full sequence of stacks and the realization of the stacks [26].

Next to the aforementioned approaches for end-to-end data-to-text generation, another important line of work in pre-neural statistical NLG has investigated models for realizing and linearizing a given hierarchical meaning representation or syntactic structure,

e.g., as part of the surface realization challenges [28,29]. Here, the most successful systems adopted statistical linearization techniques. For instance, the system by Bohnet et al. [30] and Bohnet et al. [31] were trained to map trees to output sequences using a series of classification and realization models. These linearization decisions are implemented as decoding procedures via beam search. More recent work on linearization has typically adopted AMR-based meaning representations and used different translation or transduction models to map these to output sentences, using decoding mechanisms from phrase-based MT systems [32,33].

Thus, implementations of decoding algorithms in NLG have often been based on general search algorithms or algorithms developed in the area of MT. In research on MT, different decoding mechanisms have been explored and described in great detail for a range of alignment-based or phrase-based systems [25,34,35]. For instance, the well-known beam search decoder implemented in the Pharaoh system [35] operates on a phrase table that aligns words or phrases in an input sentence with different translation candidates in the output sentence. The decoding problem is to find a high-scoring combination of translation hypotheses and, at the same time, reduce the search space which grows exponentially with the length of the input sentence. Other work on decoding in MT has investigated methods for exact inference or optimal decoding aimed at finding the best possible translation in the huge space of candidates [36,37].

### 2.2. Neural Autoregressive NLG

The NLG systems described in Section 2.1 explored a variety of computational approaches for modeling language generation with statistical methods while, importantly, assuming some form of underlying structure or linguistic representation of the sequence to be generated. Recent work in the area has focused to a large extent on the neural generation paradigm where the sequence generation task is framed as a conditional language modeling problem. Neural generation architectures are frequently called "encoder-decoder architectures" as they first encode the input $x$ into some hidden, continuous representation and then decode this representation to some linguistic output in a word-by-word manner. It is important to note the difference between the term "decoder" which refers to a part of the neural model that maps the encoded input to word probabilities and the "decoding procedure" which is an algorithm external to the model applied during inference. This survey focuses on the latter type of decoding.

Neural text generation systems generally assume that the output text is a flat sequence of words (or symbols, more generally) drawn from a fixed vocabulary $V$. The probability of a sequence over this vocabulary can be factorized into conditional word probabilities. More specifically, the probability of a word $y_j$ (to be generated at the position $j$ in the sequence) is conditioned on some input $x$ and the preceding words of the sequence:

$$\log P(y|x) = \sum_{j=1}^{J} \log P(y_j|y_1^{j-1}, x). \tag{1}$$

Such a generation model assigns a probability to all potential sequences over the given vocabulary $V$, i.e., it scores every $y, y \in V^*$, which is the main idea underlying traditional and neural language models. The word probabilities are typically conditioned on an input $x$ that is given in training data, e.g., some database record, a meaning representation or an image. In the case of so-called open text generation, the input can be empty and the formula in Equation (1) is identical to a language modeling process. In addition, in auto-regressive sequence generation, each word is conditioned on the word generated at the previous time step.

There are several neural network architectures that can be used to implement sequence generation systems as defined in Equation (1). A common example is recurrent neural networks (RNNs) that are able to consume or encode input sequences of arbitrary length and transform them into output sequences of arbitrary length [38,39]. The main idea of

RNNs is to learn to represent the hidden states of the sequence, i.e., $h$, which represents a sort of memory that encodes preceding words in the sequence:

$$\log P(y|x) = \sum_{j=1}^{J} \log P(y_j|y_1^{j-1}, x, h). \tag{2}$$

In a simple recurrent architecture, the processing of a sequence is implemented, at least, with the following hidden layers:

$$\begin{aligned} h_t &= \sigma(W^{hx}x + W^{hh}h_{t-1}) \\ y_t &= \text{softmax}(W^{yh}h_t) \end{aligned}.$$

An important limitation of RNNs is that they process both the input and the output in a strict left-to-right fashion and make it difficult to pass information between the encoder and the decoder in a flexible way. Therefore, the transformer architecture by Vaswani et al. [40] has now replaced RNNs in many neural generation settings. The central element of the transformer are self-attention heads. The layers of the transformer are built out of many such attention heads which operate in parallel. The self-attention in the encoder is not directional, as the attention at a particular position can attend to all other positions in the input sequence. The decoder of the transformer is most often implemented in an autoregressive fashion, masking out all positions following the current position. The decoder can attend to all positions up to the current one via self-attention, and includes encoder-decoder attention layers that allow the decoder to attend to all positions in the input sequence. Thus, most, but not all, neural generation systems are autoregressive, see the next Section 2.3 for a brief summary of non-autoregressive approaches in generation.

While neural autoregressive NLG models generally model the probability of a sequence as a sequence conditional word probabilities, there are different ways in which these models can be optimized during training. A standard approach is to train in a supervised manner and maximize the likelihood of a word by minimizing the cross-entropy loss between predicted tokens and tokens given in the training examples. This training regime entails that the training signal is given to the model only on the word level. A popular alternative to word-level supervised training are methods from reinforcement learning (RL) which make it possible to give sequence-level rewards to the model [41–45]. A well-known example is Ranzato et al. [41]'s self-critical sequence training used to optimize RNNs, as a variant of the REINFORCE policy gradient optimization algorithm [46]. In this approach, the prediction of the next word corresponds to an action which updates the state of the RL environment (here, the hidden state of an RNN). Importantly, the model receives the reward at the end of the sequence, such that the reward represents a sequence-level goal or quality criterion. A common reward function is the BLEU metric [47], which is also frequently used for automatic evaluation of generated sequences. It is important to note, however, that training a neural sequence generation from scratch using only RL-based sequence-level rewards is not deemed feasible in practice. In Ranzato et al. [41] and other RL-based training regimes, the generation model is pre-trained using cross-entropy loss and used as the initial policy which is then fine-tuned with sequence-level training. Section 5.3 discusses further connections between decoding and RL-based sequence-level training.

Regardless of the choice of architecture (e.g., recurrent or transformer models) and training regime (word-level or sequence-level), existing neural generation models do not provide a built-in mechanism that defines the reconstruction of the sequence from the given word probabilities. This stands in contrast to other statistical generators sketched in Section 2.1 where the sequence generation process is typically restricted by a grammar, template or tree structure. For this reason, the decoding procedure (external to the model) has a central role in the neural generation process as it needs to determine how the output sequence is assembled and retrieved from the exponentially large space of candidate sequences. Given the factorization of the sequence generation problem from Equation (1),

the decoding step needs to compose an output utterance in an incremental word-by-word fashion.

*2.3. Neural Non-Autoregressive Generation*

Shortly after the discovery of the transformer architecture by Vaswani et al. [40], researchers have started exploring the idea of parallelizing not only the encoder, but also the decoder of the neural generation architecture, leading to so-called non-autoregressive models. One of the first successful implementations of parallel decoding (here understood as the decoder part of the model) was the WaveNext architecture for text-to-speech synthesis by Oord et al. [48]. Gu et al. [49] proposed a model for non-autoregressive machine translation, with the aim of fully leveraging the performance advantage of the Transformer architecture during decoding and avoid slow, potentially error-prone decoding mechanisms, such as beam search. The main idea of non-autoregressive modeling is that, at inference time, the model does not have to take into account dependencies between different positions in the output, such as this naive baseline:

$$\log P(y|x) = \log p_L(T|x_{1:T'}) + \sum_{j=1}^{T} \log P(y_t|x_1^{T'}). \tag{3}$$

This simple non-autoregressive model predicts the target length of the sentence from its input and conditions the word probabilities only on the input, not on preceding output words. This, unsurprisingly, has not been found to work in practice as this model exhibits full conditional independence. Generally, attempts at implementing non-autoregressive models, to date, have been more or less successful. Most studies show that non-autoregressive models typically generate output of lower quality then outputs of autoregressive models. However, they are much faster and in some domains, such as speech synthesis or machine translation, good quality can be reached by using techniques of knowledge distillation [49], probability density distillation [48], or iterative refinement [50].

Besides speeding up conventional procedures for decoding in autoregressive generation, some work on non-autoregressive or partially autoregressive models aims at going beyond the assumption that output needs to be produced in a fixed left-to-right generation order. Gu et al. [51] present a transformer that treats generation order as a latent variable in sequence generation. They train their transformer to predict the next word and, based on the next word, the next position in a given partial sequence. Since the learning of a model that optimizes the likelihood marginalized over generation orders is intractable, they approximate the latent generation orders using beam search to explore the space of all permutations of the target sequence. In a similar vein, Stern et al. [52] develop the Insertion Transfomer which is trained to predict insertions of words into a partial sequence. By adopting different loss functions, their model can accommodate different generation orders, including orders that can be parallelized (e.g., balanced binary trees). Both Gu et al. [51]'s and Stern et al. [52]'s experiments show that insertion-based decoding models reach state-of-the-art performance in tasks, such as MT, code generation, or image captioning.

Generally, the design of non-autoregressive models typically involves a built-in mechanism that defines the assembly of the sequence, in contrast to the autoregressive generation models discussed in Section 2.2. For instance, the Insertion Transformer by Stern et al. [52] explicitly learns operations that manage the construction of the sequence generation, whereas these operations would be handled by the decoding method in autoregressive generation. Hence, this survey focuses on decoding methods for autoregressive generation.

*2.4. Summary*

The brief summary of some pre-neural statistical NLG systems in Section 2.1 has shown that decoding mechanisms have always played a certain role in statistical generation

systems: except the early generate-and-rank architectures where classifiers where used to score full sentences, subsequent systems have generally decomposed the generation process into smaller decisions that can be learned from a corpus, e.g., the selection of the next slot from a meaning representation or database record, the prediction of the next word, or the ordering of two words in a tree, etc. This decomposition entails that most statistical generation frameworks deal with a large number of potential output sequences. Handling this search space over generation outputs has been investigated and tackled in some pre-neural systems, especially in early end-to-end systems for data-to-text generation. Here, a couple of elaborate decoding methods have been proposed as, e.g., in Konstas and Lapata [24]'s or Mairesse and Young [26]'s work. However, there has been little effort in pre-neural NLG on generalizing these methods across different frameworks, apart from research on decoding in MT which has explicitly studied the effect of different decoding procedures. The recent neural encoder-decoder framework, introduced briefly in Sections 2.2 and 2.3, has led to NLG models that do not impose any hard constraints or structures controlling how word-level predictions should be combined in a sequence. Handling the search space in neural generation, therefore, becomes a real challenge: exhaustive search is intractable and simple search (e.g., greedy decoding) does not seem to work well in practice.

In short, the main points discussed in Section 2 can be summarized as:

- research on decoding in non-neural frameworks based on structured search spaces (e.g., hypergraphs, factored language models),
- autoregressive (left-to-right) neural NLG generally requires a decoding method defining the assembly of the sequence, and
- non-autoregressive generation methods are faster and define operations for assembling the sequences as part of the model, but often perform worse than autoregressive approaches.

## 3. Decoding as Search for the Optimal Sequence

The most widely used, and debated, decoding algorithm in various sequence-to-sequence frameworks to date is beam search, a basic breadth-first search algorithm. Many of the more advanced or specialized decoding methods discussed below build upon beam search or aim to address its limitations. In the following, we will generally introduce decoding as a search problem (Section 3.1) and discuss a basic example of beam search (Section 3.2). Section 3.3 surveys different variants and parameters of beam search used in the recent NLG and MT literature. Section 3.4 summarizes the recent debate about strengths and weaknesses of beam search, and Section 3.5 concludes with an overview of how beam search is used in practice. Table 1 summarizes these various aspects and papers related to beam search discussed in Section 3.

### 3.1. Decoding as Search

In neural NLG, decoding is most commonly viewed as a search problem, where the task is to find the most likely utterance $y$ for a given input $x$:

$$\hat{y} = \arg \max_{y \in V^*} P(y|x). \tag{4}$$

A principled approach to solving this equation would be exact search over the entire search space. This is typically unfeasible given the large vocabulary that neural generators are trained on and the long sequences they are tasked to generate (i.e., sentences or even entire texts). More formally, Chen et al. [53] prove that finding the best string of polynomial length in an RNN is NP-complete.

When viewed from a search perspective, the objective of decoding is to generate an output text that is as close as possible to the optimal output that could be found with exhaustive search. The simplest way to approximate the likelihood objective is to generate the most likely word at each time step, until an end symbol has been generated or the

maximal number of time steps has been reached. This greedy search represents a rather naive approach as it optimizes the probability of the sequence in an entirely local way. Consequently, it has been shown to produce repetitive or invariable sentences [54]. A more widely used way of approximating exact search in decoding is beam search. The discovery of this algorithm is attributed to Lowerre [8]. The main ideas, variants, and shortcomings of this algorithm will be discussed in the following.

**Table 1.** Overview of papers on beam search reviewed in Section 3.

|  | **Papers** |
| --- | --- |
| **Standard definitions** | Lowerre [8], Graves [39], Klein et al. [55], Stahlberg and Byrne [56], Meister et al. [57] |
| **Variants** | |
| stopping criteria | Klein et al. [55], Huang et al. [58], Newman et al. [59] |
| length normalization | Graves [39], Klein et al. [55] |
| length reward | Huang et al. [58], He et al. [60], Murray and Chiang [61] |
| shrinking beam | Bahdanau et al. [62] |
| coverage | Klein et al. [55] |
| pruning thresholds | Freitag and Al-Onaizan [63] |
| formal reformulations | Rush et al. [37], Meister et al. [57] |
| **Analyses** | |
| negative effect of large beam width | Yang et al. [64], Koehn and Knowles [65], Cohen and Beck [66] |
| positive effect of large beam width | Vinyals et al. [67], Karpathy and Fei-Fei [68] |
| bias towards short sequences | Graves [39], Huang et al. [58], Murray and Chiang [61], Sountsov and Sarawagi [69], Zarrieß and Schlangen [70], Newman et al. [59] |
| repetitive output | Karpathy and Fei-Fei [68], Li et al. [54], Holtzman et al. [71] |
| bias towards same prefix hypotheses | Freitag and Al-Onaizan [63], Shao et al. [72], Kulikov et al. [73] |
| usefulness of beam search objective | Stahlberg and Byrne [56], Meister et al. [74] |

### 3.2. Beam Search: Basic Example

Beam search is a pruned version of breadth-first search that keeps a fixed number of candidates on the beam, at each step of the search procedure. It can be implemented as a graph-based algorithm that builds up a tree of output sequences by incrementally adding words to the high-scoring candidates on the beam, as shown in Figure 1. The key parameter in beam search is the beam width $k$ which determines the number of candidates that will be kept at each time step or level of the tree. Each (partial) output sequence or hypothesis is associated with a score, e.g., the probability assigned to it by the underlying neural language model. All hypotheses that have a score lower than the top-$k$ candidate are pruned. Beam search with $k = 1$ is identical to greedy search which generates a single hypothesis with the most probable word given the previous words at each time step. Beam search with an infinite beam amounts to full breadth-first search. For a formal definition of the algorithm, see the pseudo code in Algorithm 1, taken from Graves [39].

The graph-based visualization in Figure 1 illustrates an example search with $k = 5$: first, the root of the tree, the start symbol, is expanded with the five most likely words that can follow the start symbol. In the second step, one of these paths (*(start)-(unk)*) is abandoned, while another path splits into 2 hypotheses (*start-in-diesem, start-in-der*). The hypothesis that is most likely at time step 2 (*(start)-(das)-(ist)*) is abanoned at time step 4. The reason for this is that one of the candidates from time step 3 (*(start)-(die)-(Kommission)-(ist)*) leads to 4 high-scoring candidates at time step 4. One of these is the final output candidate (*(start)-(die)-(Kommission)-(ist)-(geschlossen)-(.)-(<end>)*) as it contains the end symbol and achieves the best score.

**Algorithm 1:** Beam search as defined by Graves [39].

**Initialise:** $B = \{\varnothing\}; Pr(\varnothing) = 1$
**for** $t = 1$ **to** $T$ **do**
 $A = B$;
 $B = \{\}$;
 **for** $y \in A$ **do**
  $Pr(y)+ = \sum_{\hat{y} \in pref(y) \cap A} Pr(\hat{y})Pr(y|\hat{y}, t)$
 **end**
 **while** *B contains less than W elements more probable than the most probable in A*
 **do**
  $y^* = $ most probable in $A$;
  remove $y^*$ from $A$;
  $Pr(y^*) = Pr(y^*)Pr(\varnothing|y, t)$;
  Add $y^*$ to $B$;
  **for** $k \in Y$ **do**
   $Pr(y^* + k) = Pr(y^*)Pr(k|y^*, t)$;
   Add $y^* + k$ to $A$;
  **end**
 **end**
 Remove all but the $W$most probable from $B$;
**end**
**Return:** $y$ with highest $logPr(y)/|y| \in B$

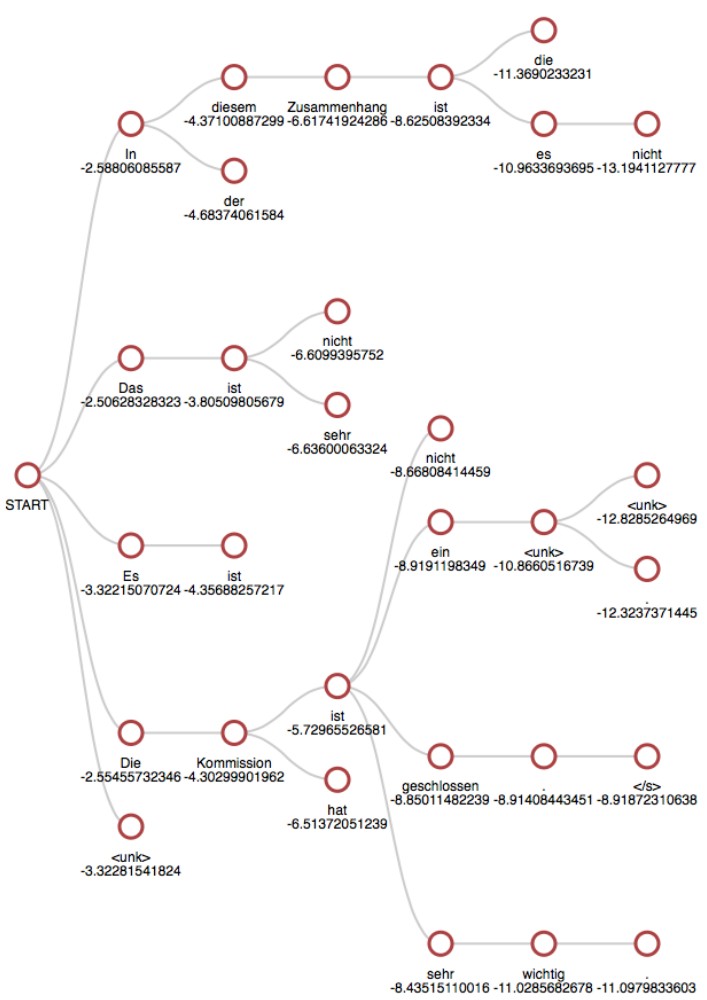

**Figure 1.** Visualization of beam search, example taken from OpenNMT.

### 3.3. Variants of Beam Search

It is important to note that beam search is a heuristic that can be defined and parametrized in different ways, beyond the width parameter $k$. Nevertheless, beam search is rarely explicitly defined in research papers. Algorithms 1–3 show three definitions of beam search from the recent literature. Interestingly, they are referred to as "standard" beam search in the respective papers but still show slightly different configurations for search: Algorithms 1 and 2 both run for a fixed number of time steps, whereas Algorithm 3 terminates once the top-most candidate on the beam ends with the special end-of-sequence symbol. Algorithms 2 and 3 do not expand hypotheses that end in an end-of-sequence symbol, whereas Algorithm 1 does not account for this case. Algorithm 1 includes a length normalization in the final selection step (last line), Algorithm 2 uses a generic scoring function which might include normalization or not, Algorithm 3 does not consider length normalization, etc.

---

**Algorithm 2:** Beam search as defined by Meister et al. [57].

**Input:** $\mathbf{x}$ : source sentence, $k$ : maximum beam size,
      $n_{max}$ : maximum hypothesis length, $score(\cdot, \cdot)$ : scoring function
$B_0 \leftarrow \{\langle 0, \text{BOS} \rangle\}$
**for** $t \in \{1, ..., n_{max} - 1\}$ **do**
    $B \leftarrow \varnothing$
    **for** $\langle s, \mathbf{y} \rangle \in B_{t-1}$ **do**
        **if** $\mathbf{y}.last() = EOS$ **then**
            $B.\text{add}(\langle s, \mathbf{y} \rangle)$
        **end**
        **for** $y \in V$ **do**
            $s \leftarrow \text{score}(\mathbf{x}, \mathbf{y} \circ y)$
            $B.\text{add}(\langle s, \mathbf{y} \circ y \rangle)$
        **end**
    **end**
    $B_t \leftarrow B.\text{top}(k)$
**end**
**Return:** $B.\max()$

---

**Algorithm 3:** Beam search as defined by Stahlberg and Byrne [56].

**Input:** $\mathbf{x}$ : source sentence, $n$ : beam size
$H_{cur} \leftarrow \{\langle \epsilon, 0.0 \rangle\}$
**repeat**
    $H_{next} \leftarrow \varnothing$
    **for** $y, p \in H_{cur}$ **do**
        **if** $y_{|y|} = \langle s \rangle$ **then**
            $H_{next} \leftarrow H_{next} \cup \{(\mathbf{y}, p)\}$
        **else**
            $H_{next} \leftarrow H_{next} \cup \bigcup_{w \in T}(\mathbf{y} \cdot w, p + \log P(w|\mathbf{x}, \mathbf{y})))$
        **end**
    **end**
    $H_{cur} \leftarrow \{(\mathbf{y}, p) \in H_{next} : |\{(\mathbf{y}', p') \in H_{next} : p' > p\}| < n\}$
    $(\tilde{\mathbf{y}}, \tilde{p}) \leftarrow \arg\max_{(\mathbf{y}, p) \in H_{cur}} p$
**until** $\tilde{y}_{|\tilde{\mathbf{y}}|} = \langle /s \rangle$;
**Return:** $\tilde{\mathbf{y}}$

---

Different usages of beam search have been discussed for a long time in the MT community, where it was the standard method in non-neural syntax- and phrase-based models [37]. In these classical phrase-based MT systems, however, candidates were all completed in the same number of steps, whereas sequence-to-sequence models generate

hypotheses of different length. Thus, beam search in neural generation requires a good stopping criterion for search or some way of normalizing scores between candidates in order to avoid a bias towards short generation outputs [39,58].

A generic solution for dealing with candidates of different length is shown in the algorithm by Graves [39] in Algorithm 1: the search terminates after a predefined number of time steps and scores for candidates are simply divided by their length. In practice, however, many neural generation systems have adopted more elaborate normalization and stopping criteria. The MT toolkit OpenNMT [55] provides a widely used implementation of beam search and includes three metrics for normalizing the coverage, length and end of sentence of candidate translations. The length penalty $lp$ is defined as follows:

$$lp(Y) = \frac{(5 + |Y|)^\alpha}{(5 + 1)^\alpha}, \tag{5}$$

with $Y$ as the current target length, and $\alpha$ as a hyperparameter that needs to be set. The length penalty is used in combination with an end of sentence penalty $ep$:

$$ep(X, Y) = \gamma \frac{|X|}{|Y|}, \tag{6}$$

with $\gamma$ as a further hyperparameter, and $|X|$ as the length of the source sentence. As this penalty depends on the length of the source sentence, it is not generally available in neural language generation, i.e., this penalty does not generalize to generation task beyond translation. In OpenNMT, the search stops once the top candidate obtained in the current step is completed (i.e., with an end symbol).

Another common neural MT framework [62] uses a shrinking beam where beam size is reduced each time a completed hypothesis is found, and search terminates when the beam size has reached 0.

Next to these implementations of beam search in standard MT frameworks, others have proposed to extend the beam search scoring function with a length reward, in order to avoid the inherent bias towards short hypotheses. He et al. [60] define a "word reward" a simple word reward as

$$s'(e) = s(e) + \gamma m, \tag{7}$$

with $s$ as the scoring function, $e$ as the hypothesis, $m$ as the hypothesis length, and $\gamma$ as the scaling factor for the reward. He et al. [60] evaluate this reward along with a features of a neural MT system and show that is beneficial in their framework. Murray and Chiang [61] present a way of tuning the simple word reward introduced by He et al. [60] and compare it to other length normalization procedures. They find that a simple word reward works well and that a tuned word reward generally performs best in their MT experiment. Huang et al. [58] introduce a related variant of beam search that is guaranteed to finish in an optimal number of steps, given a particular beam size and combine this with a "bounded length reward" that rewards each word, until an estimated optimal output length has been reached. They show that their decoding method outperforms OpenMNT's implementation of beam search and the shrinking beam in Bahdanau et al. [62].

In sequence-to-sequence generation beyond MT, not a lot of work has been done on defining general stopping and normalization criteria. Zarrieß and Schlangen [70] present a study on decoding in referring expression generation (REG), a relatively constrained NLG sub-task where the length of the generated output is deemed central. They find that a variant of beam search that only keeps hypotheses if the same length, i.e., discards complete hypotheses that are not top candidates in the current time step, provides a better stopping criterion for REG than other criteria that have been explored in the MT literature.

Freitag and Al-Onaizan [63] extend the idea of a dynamic beam width in Bahdanau et al. [62]'s shrinking beam and implement pruning of candidates from the beam that are far away from the best candidate. They investigate different pruning schemes, i.e., relative or absolute probability thresholds for pruning and a pruning scheme that fixes the

number of candidates that are expansions of the same hypothesis. They find that pruning speeds up decoding and does not decrease translation quality.

While the above papers extend beam search with specific parameters, others have aimed at more efficient formalizations of beam search. Rush et al. [37] present a variant of beam search for syntax- and phrase-based MT that comes with guarantees a bound on the possible decoding error and is faster. Meister et al. [57] develop a generic reformulation of beam search as agenda-based best-first search. Their implementation is faster than standard implementations and is shown to return the top hypothesis the first time it encounters a complete hypothesis.

The different variants and parameters of beam search discussed in this section are summarized in Table 1.

### 3.4. Beam Search: Curse or Blessing?

The discussion of different beam search versions is directly connected to is limitations which have been widely noted in the recent literature. One of these limitations has become known as "the beam search curse" and will be discussed below, together with other issues. At the same time, some recent work has argued that beam search should be seen as a blessing as it implicitly compensates for deficiencies of neural generation models. The current section will give a summary of this debate.

In 2017, Koehn and Knowles [65] mentioned beam search as one of the six most important challenges in neural MT. One year later, Yang et al. [64] referred to this challenge as the "beam search curse": both Yang et al. [64] and Koehn and Knowles [65] showed that increasing the width of the beam does *not* increase translation quality: the quality of translations as measured by the BLEU score drops with higher values of $k$. Theoretically, this should not happen as a wider beam takes into account a larger set of candidates and, therefore, should eventually decode the optimal generation output more often than a decoder that searches a smaller space of candidate outputs.

This highly undesired and unexpected negative correlation between quality and beam width has been discussed in relation to the length bias of sequence-to-sequence models [58,61,69]. It has long been noticed that neural sequence transduction models are biased towards shorter sequences [39] and that this bias results from the fact that neural MT and other generation models build probability distributions over candidates of different lengths. Murray and Chiang [61] show that correcting the length bias with a simple word reward helps eliminating the drop in quality for wider beams, though they do not obtain better BLEU scores from wider beams with their method. Interestingly, Stern et al. [52] also note that their non-autoregressive insertion transformer obtains better performance (up to 4 points in BLEU) when using an EOS penalty, i.e., a scalar that is substracted from the log probability of the end token.

Newman et al. [59] take up the issue of stopping or generating the special EOS symbol in sequence-to-sequence models. They compare two settings: models that are trained on sequences ending in EOS (+EOS) and models trained on sequences without EOS (–EOS). They find that the –EOS models achieve better length generalization on synthetic datasets, i.e., these models are able to generate longer sequences than observed in the training set. They observe that the +EOS models unnecessarily stratify their hidden state representations by linear position in the sequence, which leads to better performance of the –EOS models. Thus, similar to the study by Stahlberg and Byrne [56], Newman et al. [59] do not attribute sub-optimal decisions in stopping to the decoding procedure, but to model design and model failure.

To date, the length bias has mostly been discussed in the MT literature, but a few studies report mixed results on the effect of beam search and beam size. Work on visual storytelling found that a larger beam size deteriorates quality of the generated stories [75]. Vinyals et al. [67] find the opposite for the decoding of their well-known image captioning model and observe a positive effect of a large beam size ($k = 20$) as opposed to a beam size of 1 (i.e., greedy search). Interestingly, in a later replication of their study, Vinyals et al. [76]

carry out further experiments with varying beam width and show that a reduction of the beam size to $k = 3$ greatly improves performances compared to $k = 20$. Karpathy and Fei-Fei [68] find that a larger beam size ($k = 7$) improves the quality of generated image descriptions but also leads to less novel descriptions being generated, a smaller beam size deteriorates quality and repeats less captions from the training set. The most comprehensive study of performance degradation caused by larger beam widths is presented by Cohen and Beck [66], who investigated this effect in MT, summarization and captioning. They find a negative effect of width on generation quality in all these tasks and explain it with so-called "discrepancies", i.e., low-probability tokens that are added to early to the beam and compensated later by high-probability tokens.

Another shortcoming of beam search observed in previous work is that the beam tends to contain many candidates that share the same (most likely) prefix [63,72,73]. The bias towards hypotheses with the same prefix is also nicely illustrated in our beam search example in Figure 1: at time step 3, the beam contains 5 hypotheses that expands 3 preceding hypotheses. At time step 4, however, the diversity of the beam is substantially reduced: 4 of the 5 candidates are expansions of a single, very probable candidate from the preceding time step. This means that a relatively high value for beam size would be needed to ensure that more diverse hypotheses that could potentially lead to more probable output are not excluded too early. This, unfortunately, contradicts other studies that report a rather detrimental effect of a large beam size. A range of works have, therefore, have looked at modifying the objective of beam search such that more diverse candidates are considered during decoding. These methods will be discussed in Section 4.

Holtzman et al. [71] observe even more dramatic weaknesses of likelihood-based decoding which they describe as the phenomenon of neural text *degeneration*: they argue that the likelihood objective used for decoding open text generation with large language models (such as GPT-2) systematically leads to degenerate text that is "generic, akward and repetitive". They find that repeated phrases incur a positive feedback loop during decoding the language model: the probability of a generating a phrases, such as, e.g., "I don't know" *increases* with *every* repetition of the phrase. In practice, this feedback loop leads to text that contains sequences of the same, likely sentence, as they qualitatively show in their paper. Therefore, Holtzman et al. [71] argue that generation models should not aim at maximizing the likelihood of the output sequence, but produce text that is *not* the most probable text. They introduce nucleus sampling which will be discussed in Section 4.

While the studies discussed up to this point emphasize the limitations of beam search, others suggest that beam search is a blessing rather than a curse, as it implicitly corrects certain built-in biases and defects of neural models. Stahlberg and Byrne [56] compare beam search to exact inference in neural MT. Interestingly, they find that the underlying MT model assigns the global best score to the empty translation in more than half of the cases, which usually not noticed as exact inference is not used for already discussed, for practical reasons. Beam search fails to find these globally optimal translations due to pruning in combination with other parameters, such as length normalization. Stahlberg and Byrne [56] interpret this as evidence for the failure of neural MT models to capture adequacy. The fact that the BLEU scores drop when decoding with a wider beam should not be blamed on beam search but on deficiencies of the model.

Meister et al. [74] follow up on Stahlberg and Byrne [56] and hypothesize that beam search incorporates a hidden inductive bias that is actually desirable in the context of text generation. They propose a more generalized way of modifying the objective of beam search and formulate regularized decoding, which adds a strategically chosen regularization term to the likelihood objective in Equation (4). They argue that beam search is implicitly biased towards a more general principle from cognitive science: the *uniform information density* (UID) hypothesis put forward by Levy and Jaeger [77]. This hypothesis states that speakers prefer utterances that distribute information uniformly across the signal. Meister et al. [74] demonstrate the connection between the UID and beam search qualitatively and test a range of regularized decoding objectives that make this explicit.

Unfortunately, they do not directly relate their observations to Holtzman et al. [71]'s observations on neural degeneration. While Meister et al. [74] argue in favor of decoding objectives that minimize the surprisal (maximize probability), Holtzman et al. [71] state that "natural language rarely remains in a high probability zone for multiple consecutive time steps, instead veering into lower-probability but more informative tokens", which seems to contradict the UID hypothesis. Thus, the debate about the merits and limitations of beam search and likelihood as a decoding objective for text generation has not reached a conclusive state in the current literature. Section 6 comes back to this general issue.

*3.5. Beam Search in Practice*

Tables 2 and 3 list a range of recent neural NLG systems for different text and data-to-text generation tasks along with their decoding strategy. These tables further corroborate some of the observations and findings summarized in this section: on the one hand, beam search is widely used and seems to be the preferred decoding strategy in most NLG tasks, ranging from translation and summarization to dialog, data-to-text generation and surface realization. On the other hand, the fact that beam search comes with a set of variants and heuristics beyond the beam widths is not generally acknowledged and potentially less well known, especially in work that does not deal with MT. Here, many papers do not report on the stopping criterion or normalization procedures, but, even in MT, the exact search parameters are not always mentioned.

The central parameter, beam width $k$ sometimes differs widely for systems that model the same task, e.g., the dialog generation system by Ghazvininejad et al. [78] uses a width of 200, whereas the system by Shuster et al. [79] uses a width of 2 (and additional trigram blocking). Some sub-areas seem to have developed common decoding conventions, e.g., in MT where advanced beam search with length and coverage penalty is common or image captioning where simple beam search versions with moderate variations of the beam width are pre-dominant. In other areas, the decoding strategies vary widely, e.g., in dialog or open-ended text generation where special tricks, such as trigram blocking, are sometimes used and sometimes not. Moreover, in these areas, beam search is often combined with other decoding strategies, such as sampling, which will be discussed below.

In short, the main points discussed in Section 3 can be summarized as:

- beam search is widely used for decoding in different areas of NLG, but many different variants do exist, and they are not generally distinguished in papers,
- many variants and parameters of beam search have been developed and analyzed exclusively for MT,
- papers on NLG systems often do not report on parameters, such as length normalization or stopping criteria, used in the experiments,
- the different variants of beam search address a number of biases found in decoding neural NLG models, e.g., the length bias, performance degradation with larger beam widths, or repetitiveness of generation output,
- there is an ongoing debate on whether some of these biases are inherent in neural generation models or whether they are weaknesses of beam search, and
- the main research gap: studies on beam search, its variants, and potentially further variants for core NLG tasks beyond MT.

**Table 2.** Neural text generation systems and their decoding settings.

| System | Search | Sampling | Hyperparameters | Eval |
|---|---|---|---|---|
| | | Machine Translation | | |
| Bahdanau et al. [62] | ✓ | - | - (shrinking beam, cf. Reference [58]) | - |
| Wu et al. [80] | ✓ | - | length penalty α 0.2, coverage penalty β 0.2, pruning, beam width 3 | ✓ |
| Sennrich et al. [81] | ✓ | - | width 12, probs normalized by sentence length | |
| Johnson et al. [82] | ? | ? | ? | ? |
| Klein et al. [55] | ✓ | - | width 5 | - |
| Vaswani et al. [83] | ✓ | - | width 4, length penalty $\alpha = 0.6$ | - |
| Ott et al. [84] | ✓ | - | width 4, length penalty $\alpha = 0.6$ | - |
| Song et al. [85] | ? | ? | ? | ? |
| Rothe et al. [10] | ✓ | - | width 4, length penalty α 0.6 | - |
| | | Summarization | | |
| See et al. [86] | ✓ | - | width 4 | - |
| Gehrmann et al. [87] | ✓ | - | length penalty, coverage penalty, repetition penalty, trigram blocking | ✓ |
| Kryściński et al. [88] | ✓ | - | trigram blocking | - |
| Narayan et al. [89] | ✓ | - | width 10 | - |
| Liu and Lapata [90] | ✓ | - | width 5, tuned length penalty, trigram blocking | - |
| Dong et al. [91] | ✓ | - | width 5, trigram blocking, tuned max. length | |
| Song et al. [85] | ✓ | - | width 5 | - |
| Radford et al. [11] | - | ✓ | top-*k* sampling, $k = 2$ | |
| Rothe et al. [10] | ? | ? | ? | ? |
| | | Dialog Response Generation | | |
| Vinyals and Le [92] | ✓ | - | greedy | - |
| Wen et al. [3] | - | ✓ | - | - |
| Serban et al. [12] | ✓ | ✓ | - | ✓ |
| Dušek and Jurčíček [93] | ✓ | - | width 20 | - |
| Li et al. [94] | ✓ | - | width 20 | - |
| Shao et al. [72] | ✓ | ✓ | stochastic beam search | ✓ |
| Das et al. [95] | ? | ? | ? | ? |
| Ghazvininejad et al. [78] | ✓ | - | width 200, max. length 30, word count penalty, likelihood penalty as in Reference [96] | - |
| Baheti et al. [97] | ✓ | - | width 20, distributional constraints | |
| Wolf et al. [98] | ✓ | ✓ | width 5 | - |
| Shuster et al. [79] | ✓ | - | width 2, trigram blocking | - |
| | | Story and Open-ended Text Generation | | |
| Fan et al. [99] | - | ✓ | top-*k* sampling, $k = 10$ | - |
| Holtzman et al. [100] | ✓ | ✓ | width 10, top-k sampling (temp. 1.8) | - |
| See et al. [101] | - | ✓ | top-*k* sampling, $1 < k < 10^5$ | ✓ |
| Zhai et al. [102] | ✓ | - | width 100 | - |
| Caccia et al. [103] | ✓ | ✓ | temperature range, stochastic beam search | ✓ |

**Table 3.** Neural data-to-text or image-to-text systems and their decoding settings.

| System | Search | Sampling | Hyperparameters | Eval |
|---|---|---|---|---|
| | | Data-To-Text Generation | | |
| Kiddon et al. [104] | ✓ | - | width 10, custom candidate selection (checklist) | - |
| Wiseman et al. [105] | ✓ | - | width 1/5 | ✓ |
| Puzikov and Gurevych [106] | ✓ | - | custom candidate selection | - |
| Gehrmann et al. [107] | ✓ | - | width 10, length and coverage penalty, custom repetition blocking | - |
| Marcheggiani and Perez-Beltrachini [108] | ? | ? | ? | ? |
| Puduppully et al. [109] | ✓ | - | width 5 | - |
| Kale and Rastogi [110] | ✓ | - | width 1 | - |
| Zhao et al. [111] | ? | ? | ? | ? |
| | | Image Captioning | | |
| Vinyals et al. [67] | ✓ | - | width 20 | ✓ |
| Xu et al. [112] | ? | ? | ? | ? |
| Karpathy and Fei-Fei [68] | ✓ | - | width 7 | ✓ |
| Rennie et al. [42] | ✓ | - | width 1/3 | ✓ |
| Lu et al. [113] | ✓ | - | width 3 | - |
| Anderson et al. [114] | ? | ? | ? | ? |
| Cornia et al. [115] | ✓ | - | - | - |
| Ippolito et al. [116] | ✓ | ✓ | diverse settings | ✓ |
| | | Referring Expression Generation | | |
| Yu et al. [117] | ✓ | - | - | |
| Castro Ferreira et al. [5] | ✓ | - | tuned with between 1 and 5, length normalization ($\alpha = 0.6$) | |
| Zarrieß and Schlangen [70] | ✓ | - | diverse settings | ✓ |
| Panagiaris et al. [118] | ✓ | ✓ | diverse settings | ✓ |
| | | Image or Video Paragraph Generation | | |
| Yu et al. [119] | ✓ | - | customized stopping criterion | - |
| Krause et al. [120] | ✓ | ✓ | 1st sentence beam, then sampling (baseline) | - |
| Krishna et al. [121] | ✓ | ✓ | width 5 | |
| Melas-Kyriazi et al. [122] | ✓ | - | repetition penalty, trigram blocking | ✓ |
| Chatterjee and Schwing [123] | - | ✓ | - | |
| Wang et al. [124] | ✓ | - | - | - |
| Salvador et al. [125] | - | ✓ | - | - |
| | | AMR-To-Text Generation | | |
| Song et al. [126] | ✓ | - | width 6 | - |
| Wang et al. [127] | ✓ | - | width 6 | - |
| Mager et al. [128] | ✓ | ✓ | width 5/10/15, nucleus sampling | ✓ |

## 4. Decoding Diverse Sets of Sequences

The previous section described decoding from the perspective of search for the optimal or a highly probable generation output. We have seen, however, that maximizing the likelihood objective during decoding has negative effects on certain linguistic properties of the output: generation outputs tend to be short and lack what is often called "linguistic diversity" [96,116,118]. Research on achieving and analyzing diversity has become an important trend in the recent literature on neural NLG, and it is often investigated in the context of decoding. In the following, we will first discuss various definitions and evaluation methods for assessing diversity of generation output (Section 4.1) We then provides an overview of methods that aim at achieving different types of diversity, which can be broadly categorized into methods that diversify beam search (Section 4.2) and sampling-based methods (Section 4.3). Despite important conceptual and technical differences between these methods, they generally adopt a view on decoding that is directly complementary to

the view of decoding as search: rather than deriving a single, highly probable generation output, the goal is to produce varied sets of outputs. Indeed, the discussion in Section 4.4 will show that there is an often observed trade-off between quality (which is optimized by search-based decoding) and diversity. Table 4 shows an overview of the paper reviewed in this section.

### 4.1. Definition and Evaluation of Diversity

Diversity or variation has always been a central concern in research on NLG (cf. Reference [2]). It was one of the major challenges for traditional rule-based systems [22,129], and it remains a vexing problem, even in state-of-the-art neural NLG systems [96,130–132]. Generally, the need for diverse output in NLG can arise for very different reasons and in very different tasks, e.g., controlling register and style in documents [133], generating entertaining responses in chit-chat dialogs [96], generating responses with certain personality traits [27], or accounting for variation in referring expressions [118,134,135] or image captioning [122,136–139]. Given the widespread interest in diversity, it is not surprising that many different definitions and assumptions on what diversity in NLG is exist in the literature. Moreover, the issue of diversity is closely linked to evaluation of NLG systems, which is generally considered one of the big challenges in the field [140–144]. Importantly, different notions of diversity adopted in work on NLG are not to be confused with "diversity" investigated in linguistics, where the term often refers to typological diversity across different languages as, e.g., in Nichols [145].

One common thread in the generation literature on diversity is to go beyond evaluating systems only in terms of the quality of the top, single-best generation output. Instead, evaluation should also take into account the quality and the diversity of the *n*-best list, i.e., a set of generation candidates for a single input. This amounts to the notion of *local diversity* (in contrast to global diversity discussed below, meaning that a generation system should be able to produce different words and sentences for the same input. Another common thread is that generation outputs should be diverse when looking globally at the outputs produced by the system for a dataset or set of inputs. Thus, *global diversity* means that the generation system should produce different outputs for different inputs.

An early investigation into local diversity is carried out by Gimpel et al. [146], who argues that MT systems should aim at producing a diverse set of candidates on the *n*-best list, in order to help users inspect and interact with the system in the case of imperfect translations. They conduct a post-editing study where human participants are asked to correct the output of an MT system and find that editors benefit from diverse n-best list when the quality of the top translation is low (they do not benefit, however, when the top translation is of high quality). Similar definitions of local diversity of have been taken up in neural generation, as for instance, in Vijayakumar et al. [147] and Li et al. [54] (see Section 4.2 for further discussion).

Local diversity can be assessed straightforwardly by means of automatic evaluation metrics. Ippolito et al. [116] present a systematic comparison of different decoding methods in open-ended dialog generation and image captioning and assess them in terms of local diversity. They use perplexity over the top 10 generation candidates for an input and the Dist-*k* measure by Li et al. [54], which is the total number of distinct k-grams divided by the total number of tokens produced in all the candidates for an input. Additionally, they include the Ent-*k* measure introduced by Zhang et al. [148] that takes into account the entropy of the distribution of n-grams in the top candidates.

A complementary view on diversity is proposed by van Miltenburg et al. [132], who analyze the *global* diversity of image captioning systems which they define as the ability of the generation system to use many different words from the vocabulary it is trained on. The main challenge here is that this vocabulary will usually have a Zipfian distribution. A system that generates globally diverse output will, therefore, need to have the ability to generate rare words from the long tail of the distribution. van Miltenburg et al. [132] test a range of metrics for quantitatively measuring global diversity: average sentence length,

number of types in the output vocabulary, type-token ratio, and the percentage of novel descriptions. Their general finding is that most image captioning systems from the year 2018 or earlier achieved a low global diversity.

The distinction between local and global diversity is not always clear-cut or, at least, not always made explicit in the reported evaluation. Another way to measure diversity that seems to have been proposed independently in different papers is a variant of the BLEU, which is typically used to score the overlap between human references and generated sentences. In the context of diversity, BLEU can also be used to score the overlap between a set of model outputs, either for a single input or an entire test set [130,149–151], where a lower self-BLEU or mBLEU would signal higher diversity.

Generally, diversity is often discussed in open-ended or creative text generation task (see discussion in Section 4.4). Here, diversity is sometimes defined in a more loose way. For instance, Zhang et al. [148] aim at building a system that generates informative and diverse responses in chit-chat dialog, where the goal is to avoid "safe and bland" responses that "average out" the sentences observed in the training set. A related view can be found in the study by Reference [152]. They view diversity as related to the model's ability to generalize beyond the training set, i.e., generate novel sentences. They argue that human evaluation, which is often seen as a gold standard evaluation is not a good way of capturing diversity as humans are not able to assess what the model has been exposed during training and whether it simply repeats sentences from the training data. Hashimoto et al. [152] propose HUSE, a score that combines automatic and human evaluation, and it can be decomposed into HUSE-D for diversity and HUSE-Q for quality.

**Table 4.** Overview of papers on diversity-oriented decoding reviewed in Section 4.

|  | **Papers** |
|---|---|
| **Evaluation approaches** | |
| local diversity | Gimpel et al. [146], Vijayakumar et al. [147], Li et al. [96], Ippolito et al. [116], Zhang et al. [153] |
| global diversity | van Miltenburg et al. [132] |
| BLEU-based | Shetty et al. [130], Wang et al. [149], Zhu et al. [150], Alihosseini et al. [151] |
| other | Hashimoto et al. [152], Zhang et al. [148] |
| **Methods** | |
| Diversified beam search | Li et al. [54], Freitag and Al-Onaizan [63], Kulikov et al. [73], Ippolito et al. [116], Kriz et al. [154], Tam [155], Melas-Kyriazi et al. [122], Hotate et al. [156] |
| Sampling | Ackley et al. [157], Holtzman et al. [71], Fan et al. [99], Caccia et al. [103] |
| Combined search and sampling | Caccia et al. [103], Massarelli et al. [158] |
| **Analyses** | |
| quality-diversity trade-off | Ippolito et al. [116], Panagiaris et al. [118], Mager et al. [128], Schüz et al. [159], Zhang et al. [160] |
| verifiability-diversity trade-off | Massarelli et al. [158] |

### 4.2. Diversifying Beam Search

As discussed in Section 3.4, a common problem with beam search is that the number of candidates explored by beam search is small, and these candidates are often similar to each other, i.e., are expansions of the same candidate from the previous step of beam search. Hence, beam search is generally not a good choice when local diversity is a target for decoding. In the literature, a whole range of heuristics and modifications of what is often

called "standard" beam search have been proposed that all share the idea of diversifying beam search. Typically, these diverse beam search versions incorporate an additional method that scores similarities of candidates or groups beam histories to make sure that future steps of beam search expand different, diverse histories.

A simple but well-known method for diverse beam search has been proposed by Li et al. [54]. They aim at generating diverse n-best lists using beam search. They introduce a penalty that downranks candidates which have a sibling on the beam with a higher score, where a sibling is a candidate that is obtained by expanding the same hypothesis from the previous step of the search:

$$\hat{S}(Y_{t-1}^k, y_t^{k,k'}|x) = S(Y_{t-1}^k, y_t^{k,k'}|x) - \gamma k'. \tag{8}$$

In Equation (8), $y_t^{k,k'}$ is a word that is added to a hypothesis $Y_{t-1}^k$, and $k'$ denotes that ranking of $y_t^{k,k'}$ among other candidates that expand the same hypothesis. The goal of this penalty is to exclude bottom-ranked candidates among siblings and to include hypotheses that might have a slightly lower probability but increase the diversity of the candidates on the beam.

A similar heuristic is proposed for MT by Freitag and Al-Onaizan [63]: they set a threshold for the maximum number of sibling candidates that can enter the beam. This approach is independently proposed and dubbed top-*g* capping in Ippolito et al. [116] (where *g* is the threshold for candidates that can enter the beam and have the same history). A slightly more involved method to encourage diverse candidates during beam search is proposed in Vijayakumar et al. [147] for image captioning: they partition the candidates on the beam into groups. When expanding a candidate in a certain group, the scores (i.e., log probabilities) of each word are augmented with a dissimilarity term. The dissimilarity measure that is found to perform best empirically is hamming diversity which penalizes the selection of a token proportionally to the number of times it was selected in previous groups.

Kulikov et al. [73] implement iterative beam search for neural conversation modeling: they run beam search multiple times (with a fixed beam width *k*) and, for each iteration, score each hypothesis for its dissimilarity to hypotheses found in previous iterations. Tam [155] goes one step further and introduces clustered beam search. Here, similarity between candidates is determined by *K*-means clustering of hypothesis embeddings. These clusters are then used as groups in References [54,63], i.e., only the top candidates from each cluster are selected for the next step of the search. This method is designed for generation in chatbots, where standard neural generators often produce very short and generic responses. To exclude these, Tam [155] introduces a further language model threshold during decoding, filtering responses which have a language model score *above* a certain threshold. A similar idea seems to have been introduced independently for sentence simplification by Kriz et al. [154], but they cluster candidates post decoding and select the candidates nearest to the cluster centroids. Ippolito et al. [116] also experiment with post-decoding clustering (PDC) but select candidates with the highest language model score from each cluster.

Work on generating longer texts, such as, e.g., image paragraphs faces the problem that the output texts tend to contain repetitions [120]. Melas-Kyriazi et al. [122] present a model that uses self-critical sequence training to generate more diverse image paragraphs, but they need to combine this with a simple repetition penalty during decoding. Hotate et al. [156] implement diverse local beam search for grammatical error correction.

### 4.3. Sampling

An alternative way of increasing the diversity of language generation output is to frame decoding not as a search but as a sampling problem. When decoding by sampling, the generator randomly selects, at each time step, a candidate or set of candidates from the distribution predicted by the underlying NLG model. While sampling typically produces

diverse text, the obvious caveat is that, eventually, very low probability outputs are selected that might substantially decrease the overall quality and coherence of the text. Thus, while beam search naturally trade-offs diversity in favor of quality, the opposite is true for sampling.

Therefore, existing sampling procedures for neural generators do not apply pure sampling but use additional heuristics to shape or truncate the model distributions. A traditional method is temperature sampling [157] that shapes the probability distribution with a temperature $t$ and can be seen as a parameter of the softmax calculation [71]:

$$P(x_l|x_{t-1...1}) = \frac{exp(u_l/\alpha)}{\sum_{l \in V} exp(u_l/\alpha)}, \tag{9}$$

where $u_l$ are the logits of the language model for elements of the vocabulary. Temperature sampling is often used with low temperatures, i.e., $\alpha < 1$, as this skews the distribution to the high probability events. A detailed evaluation of the effect of temperature on quality and diversity is reported by Caccia et al. [103]: they find the neural language models trained with a standard MLE objective outperform GANs in terms of the quality-diversity trade-off, and temperature can be used to systematically balance this trade-off.

Furthermore, nucleus [71] and top-$k$ sampling [99] are well-known decoding methods aimed at increasing diversity. Both strategies are very similar in that they sample from truncated language model distributions: In each decoding step, a set of most probable next tokens is determined, from which one item is then randomly selected. They differ, however, in how the distribution is truncated. Top-$k$ sampling always samples from a fixed number of $k$ items. The sum of the probabilities of the top $k$ items, $p' = \sum_{x \in V(k)} P(x|x_{i<t})$, is then used as a rescaling factor to calculate the probability of a word in the top-$k$ distribution:

$$p' = \begin{cases} P(x_l|x_{t-1...1})/p' & \text{if } x \in V^{(k)} \\ 0 & \text{otherwise} \end{cases}. \tag{10}$$

Depending on the shape of the distribution at a given time step, $p'$ can vary widely, as noticed by Holtzman et al. [71]. Thus, if the distribution is very peaked, $p'$ might be close to 1; if it is flat, $p'$ might be a small value. For this reason, it might be difficult to select a value for $k$ that performs consistently for different distribution shapes throughout a generation process.

This shortcoming of top-$k$ sampling is addressed in Reference [71]'s nucleus sampling method: here, the decoding samples from the top-$p$ portion of the accumulative probability mass, where $p$ is a parameter that determines the vocabulary size of the candidate pool.

$$x \in V^{(p)}, \text{if } P(x|x_{i<t}) \geq p. \tag{11}$$

As the probability distribution changes, the candidate pool expands or shrinks dynamically. This way, nucleus sampling can effectively leverage the high probability mass and suppress the unreliable tail.

In practice, top-$k$ and nucleus sampling are often found in combination with temperature sampling. Moreover, sampling can be integrated with beam search and replace the typical likelihood scoring. Caccia et al. [103] call this procedure stochastic beam search: the width of the beam defines the number of words that are sampled for each hypothesis at each time step. Massarelli et al. [158] propose a similar method, called delayed beam search, where the first $L$ tokens of a sentence are generated via sampling, and the rest of the sentence is continued via beam search.

### 4.4. Analyses of Trade-Offs in Diversity-Oriented Decoding

Tables 2 and 3 include neural NLG systems that implement decoding strategies targeted at diversity. Generally, the sample of systems shown in these tables suggests that diversity-oriented decoding is used in practice, but is most widespread in "open"

generation task, such as story generation or dialog, and in tasks where output longer than a single sentences needs to be generated. In these NLG domains, even the first papers that implemented neural systems mentioned the need to integrate sampling or diversification to prevent the output from being unnaturally repetitive [92,99]. In the case of story generation or paragraph generation, sampling is further combined with additional constraints aimed at avoiding repetitions in long texts, such as, e.g., trigram blocking in Melas-Kyriazi et al. [122].

Among the many papers that described decoding in MT systems in Table 2, there is not a single paper that uses diversity-oriented decoding, and the same holds for data-to-text generation. In summarization, Radford et al. [11]'s system uses top-*k* sampling, but their work does not primarily aim at improving the state-of-the-art in summarization. In image captioning and referring expression generation, two studies explicitly aim at understanding the impact of diversity-oriented decoding in these tasks [116,118], whereas other systems do not seem to generally adopt them. For AMR-to-text generation, Mager et al. [128] compare search-based decoding and sampling and find that the latter clearly decreases the performance of the system.

One of the most exhaustive studies on diverse decoding is presented by Ippolito et al. [116]: they compare 10 different decoding methods, both search- and sampling-based, for the tasks of image captioning and dialog response generation. They propose a detailed evaluation using automatic measures for computing local diversity (in terms of entropy and distinct n-grams, see Section 4.1) and correlating them with human judgements of adequacy, fluency and interestingness. They observe that there generally seems to be a trade-off between quality and diversity, i.e., decoding methods that increase diversity typically do so at the expense of quality, and vice versa. Using a sum-of-ranks score over different evaluation metrics, they establish that clustered beam search and standard beam search with a relatively large beam width ($k = 100$) perform best for dialog generation. In image captioning, the sum-of-rank score favors random sampling with top-*k* sampling and PDC. The same trade-off is observed and analyzed by Panagiaris et al. [118] for referring expression generation.

Another trade-off of sampling-based diversity-oriented decoding is discussed by Massarelli et al. [158]: they investigate open text generation, where a large language models is tasked to continue a given textual prompt. They evaluate the verifiability of these freely generated texts against Wikipedia, with the help of an automatic fact-checking system. They show that sampling-based decoding decreases the repetitiveness of texts at the expense of verifiability, whereas beam search leads to more repetitive text that does, however, contain more facts that can be supported in automatic fact checking.

Finally, we observe that most of the decoding methods discussed in this section are designed to increase the local diversity of generation output. van Miltenburg et al. [132] present a study that evaluates the global diversity of image captioning systems using available generated image descriptions. They do not take into account possible effects of the systems' decoding methods. Schüz et al. [159] compare the global diversity of beam and greedy search, nucleus decoding, and further task-specific, pragmatically-motivated decoding methods in the more specific setting of discriminative image captioning; see Section 5.3.

### 4.5. Summary

This section has shown that diversity is an important issue that arises in many tasks concerned with the generation of longer or creative text, and that has been tackled in a range of recent papers. At the same time, existing methods that push the diversity of the generation output of neural systems in one way or another, i.e., by diversifying search or by sampling, seem to generally suffer from a quality-diversity trade-off. We will resume the discussion of this observation in Section 6.

In short, the main points discussed in Section 4 can be summarized as:

- different notions of diversity have been investigated in connection with decoding methods in neural NLG,
- diversity-oriented decoding methods are either based on beam search or sampling, or a combination thereof,
- analyses of diversity-oriented decoding methods show trade-offs between diversity, on the one hand, and quality or verifiability, on the other hand,
- diversity-oriented decoding is most often used in open generation tasks, such as, e.g., story generation, and
- the main research gap: studies that investigate and consolidate different notions of diversity, methods that achieve a better trade-off between quality and diversity.

## 5. Decoding with Linguistic Constraints and Conversational Goals

In the previous sections, we discussed rather domain-general decoding procedures that apply, at least theoretically, to most neural NLG systems and NLG tasks. This follows a general trend in research on neural NLG where, in recent years, systems have become more and more domain-independent and developers often refrain from building domain-specific knowledge into the architecture. In many practically-oriented or theoretically-motivated NLG systems, however, external knowledge about the task at hand, particular hard constraints on the system output, or simply linguistic knowledge on the involved phenomena *are* given at training and/or testing time. In neural NLG systems, it has become difficult to leverage such external constraints and to control them in terms of their linguistic behavior [161,162], as most of the processing is carried out on continuous representations in latent space. Thus, since decoding operates in the symbolic search space, it constitutes a natural place in the neural architecture to incorporate domain- or task-specific knowledge or reasoning and control mechanisms that target particular linguistic aspects of the generation output. This section discusses such approaches to decoding, which can be divided into methods that introduce lexical constraints (Section 5.1), constraints on the level of structure and form (Section 5.2), or pragmatic reasoning (Section 5.3). An overview of the different methods is shown in Table 5.

**Table 5.** Overview of papers on decoding with linguistic constraints (Section 5).

|  | Task | Decoding Method |
| --- | --- | --- |
| **Lexical constraints** | long text generation | beam search with simple candidate filter, Kiddon et al. [104] |
|  | data-to-text | bema search with simple candidate filter, Puzikov and Gurevych [106] |
|  | open vocabulary image captioning | constrained beam search Anderson et al. [161] |
|  | post-editing in MT | grid beam search, Hokamp and Liu [163]; grid beam search, Post and Vilar [164] |
|  | dialog generation | beam search with topical and distributional constraints, Baheti et al. [97] |
| **Structure and form** | poetry generation | beam search with simple filtering, Zhang and Lapata [165]; automaton-based decoding, Ghazvininejad et al. [166] and Hopkins and Kiela [167] |
|  | task-oriented dialog | constrained beam search, Balakrishnan et al. [162] |
| **Pragmatics** | image captioning | RSA, Andreas and Klein [168], Cohn-Gordon et al. [169]; discriminative beam search, Vedantam et al. [170] |
|  | zero-shot REG | modified RSA, Zarrieß and Schlangen [171] |
|  | data-to-text | distractor-based and reconstructor-based decoding, Shen et al. [172] |
|  | dialog generation | modified RSA, Kim et al. [173] |
|  | MT | trainable decoding, Gu et al. [174] |
|  | REG | trainable decoding, Zarrieß and Schlangen [70] |

### 5.1. Lexical Constraints

The need to incorporate lexical constraints in an NLG architecture can arise in different tasks and for different reasons. In some cases, they might be integrated as simple filters or criteria in standard beam search decoding. For instance, Kiddon et al. [104] present a neural checklist model for long text generation, where a list of agenda items is given in the input. They decode the model using beam search and select the most probable candidate which mentions most items from a given agenda. A similar "trick" is used in the data-to-text generation system by Puzikov and Gurevych [106], where they make sure that the candidate that mentions most attributes from the input representation is selected from the beam.

It is, however, not a convincing solution to generally incorporate lexical constraints at the end of search as a very large beam width could be required to produce the desired candidates. This is the case when, for instance, lexical constraints are complex and span several words or when the corresponding words are assigned low probabilities by the underlying model. Anderson et al. [161] make such a case for image captioning and use lexical constraints during decoding to extend the models coverage to a wider set of object classes. They argue that a severe limitation of image captioning systems is that they are difficult to extend and adapt to novel types of concepts and scenes that are not covered in the training data, whereas simple image taggers are easier to scale to new concepts. They develop constrained beam search, illustrated in Figure 2a, which guides the search to include members from given sets of words (external image labels, in their case) during decoding. The main idea of the algorithm is that the set of constraints is represented as a finite-state machine, where each state maintains its own beam of generation candidates. Interestingly, they observe that their constrained-based approach outperforms a competing system that uses similar knowledge to extend the training data of the system.

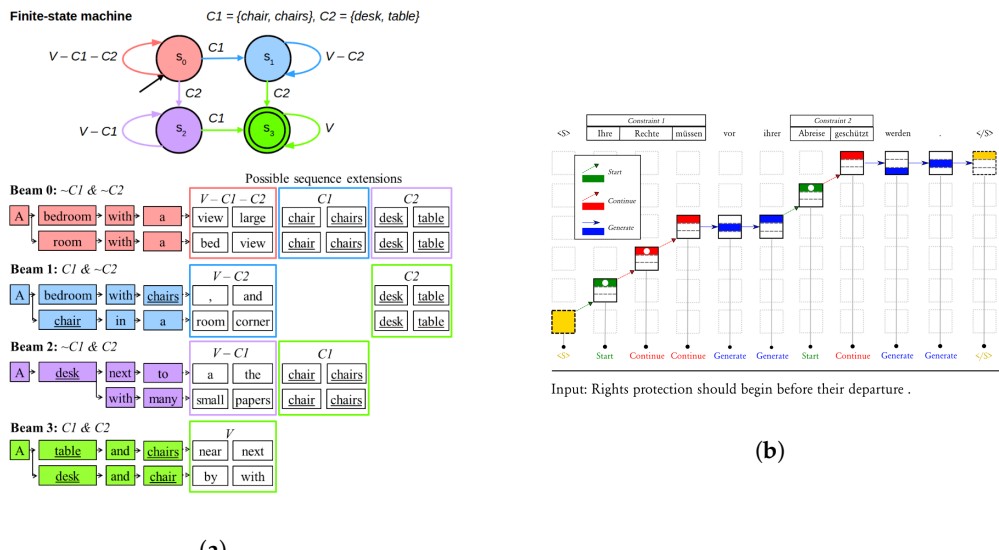

**Figure 2.** Versions of the beam search that incorporate lexical constraints. (**a**) Constrained beam search [161]. (**b**) Grid beam search [163].

A similar use case for MT is addressed in Hokamp and Liu [163], where lexical constraints are provided by users that post-edit the output of a translation system. Hokamp and Liu [163]'s grid beam search is illustrated in Figure 2b and, in comparison to the constrained beam search in Anderson et al. [161], also targets phrasal lexical constraints that span multiple words and multi-token constraints where spans might be disconnected. This beam search variant distinguishes 3 operations for expanding a candidate on the beam: open in a new hypothesis (add a word from the model's distribution), start a new constraint, and continue a constraint. For each constraint, the algorithm allocates a separate beam $B_0 B_1 ... B_C$ that groups hypotheses that meet $i$ constraints from the set. At

the end, the algorithm returns the highest scoring candidate from $B_C$, i.e., the sub-beam with hypotheses that meet all constraints. Their experiments show that grid beam search is useful for interactive post-editing and for modeling terminology in domain adaptation.

Post and Vilar [164] note that both constrained beam search and grid beam search have a high complexity that grows linearly (for grid beam search) or exponentially (for constrained beam search) with the number of constraints. They present a faster variant of grid beam search that has a global, fixed beam width and dynamically re-allocates portions of the beam to groups of candidates meeting a different number of constraints and being available at a given time-step. Their algorithm prevents the model from generating the EOS symbol unless all constraints have been met in a given candidate. In their analysis, Post and Vilar [164] take up the discussion in the MT literature, revolving around the issue of larger beam sizes resulting in lower BLEU scores by Koehn and Knowles [65] (see Section 3.4). Post and Vilar [164] observe an effect they call "aversion to references". They show that, by increasing the beam width and including partial references (i.e., constraints) during decoding, the model scores decrease, but the BLEU scores increase, which is complementary to the model scores increasing and BLEU scores decreasing in the experiment of Koehn and Knowles [65].

While the above approaches incorporated lexical constraints in a symbolic way, Baheti et al. [97] propose a distributional approach to extend the decoding procedure for a generator of conversational responses in open chit-chat dialog. They use an external topic model and an external embedding to extend the objective of standard beam search with two additional terms, scoring the topical and the semantic similarity of the source and response sentence. Furthermore, they combine this objective with the diverse-decoding method by Li et al. [54] and find that this combinations produces rich-in-content responses, according to human evaluation.

### 5.2. Structure and Form

Other interesting applications of similar decoding techniques can be found in generation tasks, where the output text does not only need to obey certain lexical-semantic constraints, but also structural and formal constraints. A prime example here is poetry generation, where systems need to produce creative text that adheres to a certain topic and, importantly, to the formal patterns of the genre, such as, e.g., rhythmic patterns, rhyme patterns, or tonal patterns [165,167]. Compared to the rather local, lexical constraints discussed in Section 5.1, these patterns are complex, need to be consistent on multiple levels (e.g., rhyme and rhythm), and need span the entire text. Nevertheless, the decoding techniques used in poetry generation systems are surprisingly similar to the one already discussed in this survey. Zhang and Lapata [165] present a recurrent neural network approach for generating Chinese poetry. Their work focuses mostly on the neural model but mentions that the tonal constraints are incorporated at the decoding stage. In their decoder, the first line generated by the RNN determines the subsequent tonal pattern, and the following lines that do not adhere to it are discarded. Ghazvininejad et al. [166] propose a similar framework for generating sonnets but focus more on the decoding stage: they construct a sophisticated finite-state automaton that represents the various formal constraints of sonnets and use the state of this FSA as additional information during beam search. Their beam search algorithm has a fixed width of 50, and they encounter the problem that this width is sometimes too narrow and does not contain words that adhere to the required rhyme patterns. Their solution is to generate the sonnet in reverse, starting from the last rhymed word. Hopkins and Kiela [167] generalize this method and compare a neural model for poetry generation, that is trained on content and formed jointly, with a model that uses a generative neural language model for generating the content of a poem with a discriminative weighted finite-state automaton that models the form of the poem. They find that the model which incorporates formal constraints in a separate discriminative model generates more formulaic poetry (e.g., makes less formal errors) and also generates poems that are rated as very human-like in a distinguishability experiment with users.

Work on incorporating structural constraints at the level of decoding is relatively scarce, when looking at tasks beyond poetry generation. A noticeable exception is the work by Balakrishnan et al. [162] presenting an approach for constrained decoding for generation in task-oriented dialog (not to be confused with the constrained beam search by Reference [161]). They tackle the often discussed issue of neural NLG systems producing semantic errors or hallucinating content in generation task, such as the E2E challenge [7], that require the accurate linguistic realization of given input data. Balakrishnan et al. [162]'s system addresses this problem by training a neural sequence-to-sequence model to not only generate sequences of words but output trees that conform to the structure of a given input meaning representation. This allows them to check, during incremental decoding of the tree structure, whether the opening bracket tokens in the output (part of the tree structure) conform to the phrases in the input representation and whether the phrase has already been included in the subtree in preceding timesteps. Their experiments show that the generated outputs are more grammatical and semantically correct as compared to systems that do not incorporate these structural constraints.

*5.3. Conversational Goals*

The aforementioned work on poetry generation has shown the idea of using a neural language model for generating the content of a message, which is subsequently refined and constrained by an external decoding method that incorporates linguistic knowledge. This idea is also spelled out in a recent line of work that aims at incorporating high-level pragmatic reasoning in the decoding of, e.g., neural image captioning systems. While standard image captioning targets (more or less) neutral descriptions of images, this task has been extended to pragmatically informative captioning in Andreas and Klein [168], Vedantam et al. [170], and Cohn-Gordon et al. [169]. These works train a neural NLG on standard image description datasets and decode this system, at testing time, to produce captions that discriminate target images from a given set of distractor images. These models are evaluated primarily in terms of their pragmatic informativeness, i.e., using a "listener" model that resolves a generated caption to an image in the context of distractor images. Generally, this setting is very similar to the well-known Referring Expression Generation (REG) task [117,175,176], except the fact that the neural generation model is trained on context-agnostic data.

The RSA framework [177] models informativity at the semantics-pragmatics interface, i.e., it provides a formalization of how pragmatically informative utterances can be derived from from literal semantics using Bayesian inference. Andreas and Klein [168] and Cohn-Gordon et al. [169] have implemented RSA as a decoding strategy which integrates pragmatic factors into the iterative unrolling of recurrent generation models. At the heart of the RSA approach is a so-called rational speaker, who reasons about how an utterance would be understood by a listener, in order to assess whether the utterance allows the identification of the target. The speaker and listener are given a set of images $W$, out of which one image $w^* \in W$ is known to the speaker as the target image.

The rational speaker in RSA is based on a *literal speaker*, who produces initial utterance candidates. In the simplest case, the literal speaker is a conditional distribution $S_0(u|w)$, which assigns equal probability to all true utterances $u \in U$ and zero probability to false utterances. The *pragmatic listener* $L_0$ then assesses the discriminative information of these candidates and is defined as follows, according to Cohn-Gordon et al. [169]:

$$L_0(w|u) \propto \frac{S_0(u|w) * P(w)}{\sum_{w' \in W} S_0(u|w') * P(w')},$$

where $P(w)$ is a prior over possible target images. The pragmatic speaker $S_1$ is defined in terms of the pragmatic listener:

$$S_1(u|w) \propto \frac{L_0(w|u)^\alpha * P(u)}{\sum_{u' \in U} L_0(w|u')^\alpha * P(u')},$$

where $P(u)$ is a uniform distribution over possible utterances $U$, and $\alpha > 0$ is a rationality parameter determining the relative influence of the pragmatic listener in the rational speaker.

The Emitter-Suppressor method (henceforth *ES*) proposed by Vedantam et al. [170] follows a similar idea as RSA decoding but is not directly grounded in pragmatic theory. ES has a less strict distinction between speakers and listeners, and it reshapes the literal predictions of the model without Bayesian inference. In ES, a speaker (*emitter*) models a caption for a target image $I_t$ in conjunction with a listener function (*suppressor*) that rates the discriminativeness the utterance with regard to a distractor image $I_d$:

$$\Delta(I_t, I_d) = \arg\max_s \sum_{\tau=1}^{T} \log \frac{p(s_\tau | s_{1:\tau-1}, I_t)}{p(s_\tau | s_{1:\tau-1}, I_d)^{1-\lambda}},$$

where $s$ is the caption for the target image $I_t$ in context of the distractor image $I_d$, and $T$ is the length of the resulting caption. $\lambda$ is a trade-off parameter that determines the weight by which $I_t$ and $I_d$ are considered in the generation of $s$. For $\lambda = 1$, the model generates $s$ with respect to $I_t$ only, thus ignoring the context. The smaller the value of $\lambda$, the more $I_d$ is weighted. In a later replication study, Schüz et al. [159] directly compared RSA and ES decoding and showed that both methods lead to broadly comparable improvements in discriminative image captioning, though there are some differences, depending on the hyperparameters and the evaluation criterion.

Zarrieß and Schlangen [171] extend RSA-based reasoning to a zero-shot setting, where the speaker's task is to refer to target object of an "unknown" category that the literal speaker has not encountered during training. This resembles the set-up described in Anderson et al. [161], where the decoding procedure extends the capabilities of the underlying language model to out-of-domain data, though Zarrieß and Schlangen [171]'s reasoning scheme does not widen the model's vocabulary but aims at leveraging the training vocabulary in efficient way for referring to unknown objects.

The aforementioned approaches deal with pragmatic reasoning in visual environments, where reasoning is based on a simple, single forced-choice task. Some extensions to non-visual tasks have been explored: Shen et al. [172] implement a model for pragmatically informative text generation, comparing so-called reconstructor-based and distractor-based reasoning schemes: in a reconstructor-based set-up, the listener predicts a distribution over all possible input contexts (e.g., meaning representations) for a generation output, whereas distractor-based reasoning scores distinguishes an input from a set of alternate, distractor inputs. Shen et al. [172]'s outperforms competitive neural generation systems without pragmatic decoding on the E2E dataset [178]. Kim et al. [173] implement pragmatic reasoning for decoding a neural dialog response generator that aims at achieving so-called public self-consciousness: the literal speaker is trained to generate responses on the PersonaChat data [153], and the listener models the identification of the speaker's underlying persona. They show that this decoding scheme improves the consistency of the generated responses, i.e., the response are less contradictory than outputs decoded without reasoning.

Up to this point, this survey has discussed decoding methods that are clearly separated from the internal layers of a neural NLG architecture and that use heuristics and algorithms to handle the symbolic search space during sequence generation. This distinction between training-modeling, on the one hand, and testing-decoding, on the other hand, is common but not always entirely clear-cut. Section 2.2 already mentioned RL-based methods for optimizing the model with sequence-level rewards in training. This sequence-level optimization aims to address the basic limitation in supervised training of standard neural generators that are optimized to achieve likelihood on the word-level, i.e., without explicit quality criteria at the sequence level. In that sense, sequence-level training is similar in motivation to some of the decoding methods discussed so far. A potential advantage of RL-based methods is that a given objective or conversational goal is not represented in a presumably heuristic algorithm that needs to be implemented anew for every goal and task but is optimized by the policy in the model. For this reason, Gu et al. [174] explore RL for

decoding and introduce the notion of trainable decoding. As in other RL-based generation approaches [41,42], they use a neural (MT) system that is trained in supervised fashion as their base model. An important difference to Ranzato et al. [41]'s approach is that, for decoding, they add an additional layer or "actor-network" to the trained model that will be optimized with RL, while freezing the other, pretrained layers of the network. They treat this actor network as a trainable decoder that learns to manipulate the hidden state of the underlying pre-trained RNN and can be optimized with any given reward function. Whereas Gu et al. [174] train the decoder actor network with a policy gradient method, Chen et al. [179] present a supervised method to train the decoder. Of course, this notion of trainable decoding is conceptually different from an actual inference procedure for sequence prediction. Thus, when applying their model, Gu et al. [174] combine the trainable decoder with the beam search heuristic. Zarrieß and Schlangen [70] test Chen et al. [179]'s supervised approach in an REG experiment and combine it with greedy search to avoid the already discussed deficiencies of beam search, while Gu et al. [174] and Zarrieß and Schlangen [70] rely on BLEU as a reward for the decoder, other metrics and rewards might constitute more interesting options to optimize decoding for, e.g., conversational goals. For instance, Panagiaris et al. [118] present a transformer-based model for REG that incorporates RL and various decoding methods to balance the diversity and informativeness of referring expressions. Their approach suggests that different objectives during generation might be achieved through a combination of modeling and decoding techniques.

*5.4. Summary*

This section has discussed a rather diverse range of generation systems and ways of integrating task-specific constraints, knowledge, or linguistic reasoning into the generation process at decoding time. A basic idea that underlies all these methods, however, is that there might be a natural division of labor between a neural generation model and a decoding methods for a given task: while the neural model can be straightforwardly trained with a likelihood objective on a large-scale training set, the decoding method can be easily set up to incorporate further constraints that extend the linguistic reasoning capabilities of the model. This division is clearly spelled out, for instance, in the RSA-based decoding in Section 5.3, where the language model represents a literal speaker, who produces likely utterances and who can be extended to a pragmatic speaker with a Bayesian decoder reasoning about informativeness in context. Contrary to the likelihood or diversity-oriented decoding procedures surveyed in Sections 3 and 4, the corresponding decoding methods are not included in Tables 2 and 3, as they are conceptually and technically diverse and often tailored to specific tasks. Here, future work might aim for a more systematic comparison with other decoding methods (see Section 6 for further elaboration on this point).

In short, the main points discussed in Section 5 can be summarized as:

- decoding methods offer themselves to be tailored to incorporate linguistic constraints at different levels of the generation process,
- decoding methods with lexical or structural constraints typically present extensions of beam search where candidates are filtered in more or less sophisticated ways,
- lexical constraints during decoding can extend a model's vocabulary,
- decoding can be used to incorporate reasoning on high-level conversational or task-related objectives, and
- the main research gap: generalize and transfer these methods across NLG tasks and develop a more systematic understanding of objectives that can be implemented at the decoding stage in NLG.

## 6. Research Gaps and Future Directions

This survey has shown that decoding methods in neural NLG can be developed and analyzed from many different perspectives and tasks. Decoding is a core part of a neural generation set-up, and many recent NLG papers have investigated its impact on generation

quality. While the recent body of work on decoding has reached robust insights into effects of certain decoding strategies, particularly beam search, a number of open questions and challenges remain to be addressed. This section discusses the main research gaps that follow from the observations made in this survey.

### 6.1. Exploring Decoding Parameters for Core NLG Tasks

A lot of research on decoding neural sequence-to-sequence models has been done in the area of MT. Most of the studies on beam search discussed in Section 3 have looked at effects of beam search exclusively in MT experiments. Consequently, many of the parameters and variants developed for beam search make assumptions that apply only, or mostly, in an MT setting. For instance, the end of sentence penalty in Klein et al. [55]'s beam search implementation assumes that the length of the output text can be estimated from the length of the input text. Missing exploration and analysis of these and further parameters for settings, such as data-to-text generation, where there is much less similarity structural between input and output, constitute an obvious research gap. Moreover, standardized implementations of flexible search strategies would be of great practical use and could support systematic evaluation and benchmarking of neural NLG.

### 6.2. The Status of Language Modeling in Neural Generation

Neural language models constitute a core part of state-of-the-art neural language generation models. As compared to pre-neural architectures, one could say that the relation between decoding and language modeling has been turned upside-down in neural NLG: pre-neural systems defined the core model based on some target representation of the output (e.g., a tree or template) and took advantage of language models during decoding, to score the likelihood and fluency of a constrained set of output candidates. In neural systems, the core model typically *is* a conditional language model scoring the likelihood of the infinite set of output texts, and decoding is used to restrict this set with some target heuristic. As language models are trained to maximize the likelihood of sequences of words observed in corpora, it follows that likelihood has become a central objective in neural NLG model. More than that, likelihood is the most well-understood objective in language generation to date as it can be achieved by the common combination of word-level supervised training and sequence-level beam search decoding. During training, it seems to be an important criterion for, at least, technical reasons. For instance, work on RL-based generation [41,42] has shown that training with more abstract sequence-level constraints is not successful.

Generally, however, the importance and status of likelihood as an objective in computational language generation does not yet seem to be well understood. This survey has shown that NLG researchers have found various weaknesses with purely likelihood-oriented generation, most notably its failures in reproducing the diversity of natural language in its various forms [54,67,71,72,92,116,131,132,147]. Therefore, Holtzman et al. [71] argues that language generation systems should produce texts that are likely but do not fully maximize likelihood, i.e., avoid consecutive, high-probability zones in text. An opposite view is taken by Meister et al. [74], who argue for the well-known principle of uniform information density from research in cognitive science. A promising method to investigate these questions is sketched by Zhang et al. [160]. They conduct a human evaluation study testing system outputs on the likelihood-diversity spectrum and confirm the likelihood trap discussed by Holtzman et al. [71]. Their results show a positive correlation between average human quality judgements and likelihood, but, importantly, they find an inflection point after which high likelihood is negatively correlated with human quality ratings.

These studies reveal that there is still a substantial research gap in the theoretical understanding of language modeling and likelihood in the training of NLG systems. An interesting direction for future work would be theoretically- and cognitively-motivated analysis and evaluation studies that deepen existing insights on the behavior of language

models in NLG tasks. This research gap also connects to a current trend in NLP on analysis methods for large neural language models (cf. Belinkov and Glass [180]).

### 6.3. Diversity, Effectiveness, and other Objectives in NLG

When speakers converse, their utterances are remarkably diverse and, at the same time, remarkably precise and effective. For instance, in a widely used corpus of human descriptions of images showing common objects, Devlin et al. [181] find that 99% of the image captions are unique. Other work that has collected such descriptions in interactive, game-based settings [182,183] found that speakers often only need a few words or utterances to unambiguously refer to objects or generally make themselves understood in a rich, communicative context. Theoretically, it is well established that speakers pursue objectives and intentions other than likelihood maximization while producing their utterances. For instance, many well-known theories on conversation and pragmatics have discovered and formulated principles of intentional and goal-oriented language use in human interaction, e.g., Grice [184] or Clark [185].

A striking result of our survey is that it is still surprisingly unclear what a good objective is, when decoding (and training) a neural NLG system. A commonly adopted solution is to relax, during decoding, the likelihood objective of the pretrained language model and sample candidate words at inference time, thereby introducing randomness into the generation process [3,71,72,98,99,116,118]. A complementary solution is to introduce constraints, knowledge or some form of task-specific reasoning into the decoding procedure, as in References [161,162,168], or to adopt some form of trainable decoding [41,174]. Except some studies that have investigated the likelihood-diversity trade-off [116,152,160], there is hardly any systematic understanding as to how these more abstract conversational goals compare to each other and between generation task. Here, Schüz et al. [159] take a first step and evaluate pragmatic reasoning in neural image captioning and compare it to diversity-oriented decoding. They show that, although not aiming at diversity itself, decoding with RSA does not only lead to more more informative utterances but also increases linguistic variation, which, in turn, leads to increased lexical diversity. This brings up the question whether certain objectives, e.g., linguistic diversity, might not need to be implemented as an explicit objective, but it could arise naturally from a more systematic understanding and implementation of conversational goals in neural NLG. A related question is on generation tasks, where multiple objectives might need to be balanced. For instance, Gkatzia et al. [186] formulate a multi-objective approach to generating summaries that aims at fulfilling the needs of different user groups of generated text. For future work, we see great potential for exploring different types of objectives in language generation, in controlled and open-ended tasks, and studying them in terms of modeling and evaluation.

### 6.4. What to Model and How to Decode?

Generally, in this survey, we have seen that so-called end-to-end neural NLG systems consist at least of the following two components: a typically complex neural model that learns to predict words conditioned on some context, and a typically less complex decoding heuristic that controls how words are strung together as sequences. In many cases, the decoding method is "innocent" in the sense that it implements the same likelihood objective as the model; in other cases, the decoding method adds substantial further assumptions, goals, and constraints to the generation process. In either case, in neural NLG architectures, there is necessary division of labor between the modeling and the decoding step which results from the simple fact that sequence probabilities need to be factorized into word probabilities.

In some papers discussed in this survey, the division of labor between modeling and decoding is made very clear. For instance, the approaches in Section 5 precisely motivate which aspect of linguistic reasoning was added to the model during decoding. However, unfortunately, this is not generally the case. The overview of systems in Tables 2 and 3 clearly suggests that there is not yet a fully standardized, systematic practice in developing

and reporting the decoding method incorporated in an NLG system. This might be, among other things, a consequence of the fact that evaluation of neural generation models is extremely challenging. Even the basic set-up of the core model involves a lot of parameters such that it is impossible to precisely test the effect of these various elements of the resulting framework, including the various parameters implemented in the decoding step. Here, we see potential in future work to establish better development and evaluation methodology that includes best practices for decoding neural NLG models. In our view, another promising direction for future work is to arrive at a more systematic understanding of the conceptual division of labor between modeling and decoding in neural NLG, i.e., which aspects of language generation should be taken care of in the model and which aspects should be handled in the decoding algorithm.

## 7. Conclusions

This article has reviewed decoding methods in neural language generation. These methods are external to the so-called end-to-end model and define an algorithm, often a more or less sophisticated heuristic, that operates in the symbolic, infinite search space of potential utterances that could be realized for a given input. Our survey was based on a broad categorization of decoding methods according to their general objective. This has shown that existing methods can be set up to optimize very different criteria and goals in the language generation process, which is both a chance and a risk: decoding can be leveraged to enhance and control the behavior of a given neural generation model that will be typically trained to predict a likely next word. At the same time, future work still needs to establish how exactly objectives, such as likelihood, diversity, or informativeness, can be integrated in a way that they complement each other, rather than leading to trade-offs and drops in quality.

**Author Contributions:** Conceptualization, S.Z. and S.S.; investigation, S.Z. and H.V. and S.S.; writing—original draft preparation, S.Z.; writing—review and editing, S.Z. and H.V and S.S.; project administration, S.Z.; funding acquisition, S.Z. All authors have read and agreed to the published version of the manuscript.

**Funding:** This research (H.V.) was partially funded by the Michael Stifel Center Jena as part of the "A Virtual Werkstatt for Digitization in the Sciences" project, funded by the Carl Zeiss Foundation (062017-02).

**Institutional Review Board Statement:** Not applicable.

**Informed Consent Statement:** Not applicable.

**Data Availability Statement:** Not applicable, the study does not report any data.

**Conflicts of Interest:** The authors declare no conflict of interest.

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
