# Peer review of "Decoding Methods in Neural Language Generation: A Survey"

_information, doi:10.3390/info12090355_

Round 1

Reviewer 1 Report

The paper represents a valuable survey on Natural Language Generation methods based on neural models.

Although it does not present any original results by its own,

the systematic completeness of the methods presented

makes the paper worthy to be published. 
The number and significance of references cited in the paper are impressive.

A few detailed comments/corrections follow:

  • line 17: nlg systems are designed, developed and learned: designed, developed and  trained with
  • line 53: the choice of decoding method: either methods or the decoding
  • line 76: with a general of: here there is a mossong noun (I suppose  overview)
  • line 103: and can designed: maybe “that can be designed “?
  • line 142: their system represents that the search: remove that
  • line 316: discussion of: on
  • line 408: but all show: subject missing, but they all
  • line 463: makes it possible a larger set: makes it possible for a 
  • line 584: lack linguistic diversity: are you really making reference  to linguistic diversity? The problem with linguistic diversity can only be related to lack of training data. I guess you would intend something else. From the ensuing lines I can see that you make reference to a “global variety” that can be perfectly understood.  However, “linguistic diversity” is a specific and well-defined research domain that is different from the one you are mentioning.
  • lines 834-837: syntax to be revised.
  • line 1130: In sone papers we discussed in this survey. ?? 

Fixed these very minor and local problems, the only main issue to deal with is the one related to language diversity which is a  research domain by itself.

Author Response

* RE: "linguistic diversity: are you really making reference to linguistic diversity? The problem with linguistic diversity can only be related to a lack of training data. I guess you would intend something else. From the ensuing lines, I can see that you make reference to a “global variety” that can be perfectly understood.  However, “linguistic diversity” is a specific and well-defined research domain that is different from the one you are mentioning."

  - We agree that there are issues with the term "diversity", as it is not well-defined in the area of NLG and used in a somewhat different sense in linguistics. Nevertheless, we'd like to stick to the term diversity here (in contrast to, e.g., variety which might fit better conceptually) as it is very commonly used in the decoding literature. We have now used quotes when introducing the term diversity `"' (l. 651) and tried to explicitly separate it from diversity in linguistics (l. 677-678).   

* RE: minor problems

  - thanks a lot, we fixed these!

Reviewer 2 Report

This manuscript aims to reviews and explore the decoding methods in the Neural Language Generation field.  The manuscript focuses on a good research direction that is interested to many researchers and readers.  

The introduction must be revised because it contains unnecessary information and fails to present the problem. Therefore, the authors should emphasize the relevance of the proposed method. However, your examination must be systematic based on a state-of-the-art study with a clear meta-analysis and synthesis (no narrative review). The review should be theoretically and critically analyzes each reference with its added value. You have to justify your paper selection and summarize your findings in a table that allows the reader a comprehensive view that guides the corresponding section. 

What are the research gaps, advantages, and disadvantages of each method?

These gaps should be of global interest and showing critical significance to the field. What did we learn compared with current, significant research?

The authors should make explicit suggestions about how their study affects the new development in the decoding techniques in the NLG field.

In addition to the following comments:

- Enhance the abstract to emphasize the objectives and the contributions of this work. And add the obtained quantitative results.

- L3-5: Too long sentence, which makes the meaning unclear. Consider breaking it into multiple sentences. Check all the text for the same comment.

L11-13: Too long and unclear sentence.

- Avoid use words such as "we," "our," etc.

-Avoid using many references together, such as L28 -L52, L102,236, etc. You should classify the studies and write a paragraph about each study or category.

L26: use the reference [2] in its correct sequence number.

-The English language, redaction, and punctuation need to be improved in general. The manuscript should undergo editing before being published. The following are some examples:

L5: In this survey paper, we provide…… should be ….   This survey paper provides

L7: real trend in the area of neural  …… should be ….   real trend of neural

L7: generation and numerous  …… should be ….     generation. Numerous

L9: The goal of this survey is to …… should be ….   This survey aims to

Author Response

* RE: "The introduction must be revised because it contains unnecessary information and fails to present the problem. Therefore, the authors should emphasize the relevance of the proposed method."

 - We have revised the intro and included a more structured overview and motivation of the survey (Section 1.1). We have also included a subsection that explains and motivates the paper selection (Section 1.2).

* RE: "However, your examination must be systematic based on a state-of-the-art study with a clear meta-analysis and synthesis (no narrative review). The review should be theoretically and critically analyzes each reference with its added value."

 - We have revised parts of Section 3, to provide a more accessible overview of different variants of beam search.
 - Sections 3, 4, and 5 included dedicated subsections where evaluation studies and analyses of the different decoding methods are discussed (Section 3.4, 4.4, 5.4). As explained in these sections, the added values of each decoding method often depend on the evaluation setting and specifics of the task.
  - But, indeed, the survey aims to provide an overview of the decoding literature in NLG in the form of a narrative review. We believe that this is a common approach to surveys in NLP, see e.g. Gatt & Krahmer's survey on NLG. Formal meta-analyses are rather difficult in NLG and NLP as the respective methods and models often differ in many aspects of the setting and in many details of the modeling.

* RE: "You have to justify your paper selection and summarize your findings in a table that allows the reader a comprehensive view that guides the corresponding section."
  - We have also included a subsection that explains and motivates the paper selection (Section 1.2).
  - In Sections 3, 4, 5, we have included Tables that provide an overview of the reviewed papers.

* RE: "What are the research gaps, advantages, and disadvantages of each method?"
  - The analysis subsections of each section (3.4, 4.4, 5.4) summarize the relevant evaluation studies and analyses of the different decoding methods. 
  - We have now included a summary of the main points at the end of each Section that, hopefully, makes the main observations more accessible.

* RE: "These gaps should be of global interest and showing critical significance to the field. What did we learn compared with current, significant research? The authors should make explicit suggestions about how their study affects the new development in the decoding techniques in the NLG field."
   - The summaries at the end of each Section now contain a brief statement of the research gaps found in each Section.
   - Section 6 contains an extensive discussion of research gaps and directions for future work in neural NLG that we identified in this survey.

* RE: "Enhance the abstract to emphasize the objectives and the contributions of this work. And add the obtained quantitative results."
  - We cannot address this comment as we do not obtain quantitative results in this survey (see comment above on the intended format of the survey).

* RE: minor comments
  - Thanks a lot for spotting these issues, we addressed these where appropriate!

Finally, thanks a lot for these extensive and constructive comments!

Round 2

Reviewer 2 Report

The author made great efforts to address my suggestions. The revised manuscript is enhanced to the level that could be published in the current form based on the editorial board's opinion.